# The evolution of human altriciality and brain development in comparative context

Aida Gómez-Robles [1,2] ✉, Christos Nicolaou [1], Jeroen B. Smaers [3] & Chet C. Sherwood [4]

Human newborns are considered altricial compared with other primates because they are relatively underdeveloped at birth. However, in a broader comparative context, other mammals are more altricial than humans. It has been proposed that altricial development evolved secondarily in humans due to obstetrical or metabolic constraints, and in association with increased brain plasticity. To explore this association, we used comparative data from 140 placental mammals to measure how altriciality evolved in humans and other species. We also estimated how changes in brain size and gestation length influenced the timing of neurodevelopment during hominin evolution. Based on our data, humans show the highest evolutionary rate to become more altricial (measured as the proportion of adult brain size at birth) across all placental mammals, but this results primarily from the pronounced postnatal enlargement of brain size rather than neonatal changes. In addition, we show that only a small number of neurodevelopmental events were shifted to the postnatal period during hominin evolution, and that they were primarily related to the myelination of certain brain pathways. These results indicate that the perception of human altriciality is mostly driven by postnatal changes, and they point to a possible association between the timing of myelination and human neuroplasticity.

Mammalian species can be classified according to their developmental patterns as altricial or precocial. Altricial species are characterized by having large litter sizes, short gestations, closed sensory organs and naked skin at birth, and by having limited mobility when they are born. Precocial species have small litters and long gestations, their sensory organs are open and functional at birth, their body hair is present when they are born, and they are able to locomote on their own soon after birth[1,2]. Human newborns are generally described as altricial because, compared with other primates, infants are underdeveloped at birth and more dependent on parental care for survival. Human altriciality, or helplessness at birth, has been associated with a high level of brain plasticity resulting in increased learning capacities, behavioural flexibility and enhanced capacity for cultural transmission[1–4]. The logic underlying the link between altricial development, brain plasticity and

behavioural complexity is that a higher proportion of brain development happening postnatally allows for neural circuitry to be shaped directly by the environment where the adult individual will have to survive, thus resulting in behaviours that are directly shaped by and adapted to those environments[4].

Although humans appear altricial when compared with other primates, a broader phylogenetic perspective that includes additional mammalian orders shows that many other species are substantially more altricial. However, although a relationship between altricial development, delayed brain maturation and enhanced learning capacities has been hypothesized for birds[5], the available evidence does not currently indicate a relationship between developmental patterns and adaptations related to social behaviour in mammals[6]. Therefore, our study aims to analyse the evolution of human altriciality relative

[1]Department of Anthropology, University College London, London, UK. [2]Department of Genetics, Evolution and Environment, University College London, London, UK. [3]Department of Anthropology, Stony Brook University, Stony Brook, NY, USA. [4]Center for the Advanced Study of Human Paleobiology, Department of Anthropology, The George Washington University, Washington, DC, USA. ✉e-mail: a.gomez-robles@ucl.ac.uk

to other primates and mammals to determine whether the evolution of our species' development departs from other groups, and whether these differences might be linked to the emergence of human specializations for brain plasticity[2,7,8].

One explanation for human altriciality is that it arose as the result of an obstetrical dilemma: as humans evolved larger brains and the birth canal became constrained due to biomechanical adaptations for bipedality, selection favoured neonates with relatively smaller brains at the time of birth, and a greater proportion of brain growth was offset to the postnatal period[9,10]. Other authors suggest that metabolic constraints are more likely to have driven the evolution of human altriciality, as foetal energetic demands towards the final part of pregnancy are too high to maintain gestation much beyond 40 weeks[11,12]. Whether human altriciality evolved because of obstetrical or metabolic factors, or a combination of both, the resulting opportunity for extrauterine brain growth and maturation may have proved selectively advantageous[2,4]. Survival of the mother and newborn during labour would be strong selective forces, but additional selective advantages in the form of increased brain plasticity and learning abilities could also explain why giving birth to underdeveloped and vulnerable infants, a seemingly disadvantageous trait, evolved in humans[1].

The altriciality–precociality spectrum is complex and multifaceted. The first part of our study, however, focuses on a simple metric, which is the proportion or percentage of adult brain size that is present at birth. This variable is broadly discussed in the human altriciality literature, as it shows a substantially lower value in humans (different studies report between 20% and 30% of adult brain size at birth[13]) than in all the other primates (35–40% in chimpanzees and even higher values in the other primates[2,4,13]). It is important to note that this percentage value provides only limited information on the level of altriciality or precociality of each species, it is not exposed to selection per se, and its study poses methodological limitations (Methods). However, it is important to understand how this proportion of neonatal to adult brain size has varied across mammalian evolution because of its historical importance in discussions about human altriciality[1,2,12,14–16]. After exploring the variation and evolution of brain and body proportion at birth across mammals, we studied the relationship between neonatal and adult brain and body sizes in a regression context.

Given the differences in developmental patterns observed between extant humans and the great apes, increased altriciality must have evolved within the hominin clade, which is the clade that includes fossil species that are more closely related to humans than to chimpanzees. However, there are uncertainties associated with the estimate of neonatal brain and body size in the hominin fossil record[17,18], which have made it difficult to infer how developmental patterns evolved in hominins. Some studies indicate that altriciality is associated with a large brain size, either through metabolic or obstetrical constraints, and that it evolved in large-brained hominins, perhaps in *Homo erectus*[4,19] (but see ref. 20). Recent research, however, suggests that a more altricial pattern of development might have evolved in earlier hominins, perhaps even in some australopiths[21,22], but these claims remain difficult to confirm or refute. Meanwhile, comparative data indicate a strongly conserved nature of neurodevelopment across all placental mammals[23], as well as an overall increase in adult brain size during hominin evolution[24–26] and probably a slight increase in gestation lengths from earlier to later hominins[27]. Based on these comparative data, it is possible to estimate of how the timing of neurodevelopment (that is, the date after conception at which each neurodevelopmental event occurs) with respect to the time of birth has changed during hominin evolution. Our analyses across the mammalian phylogeny and within the hominin clade can help us understand whether humans are unexpectedly altricial given their evolutionary context, and whether a more altricial pattern of development has been selected in humans because of its association with increased neuroplasticity.

## Results

### Brain and body proportion at birth across mammals

Brain and body proportion at birth (measured as the percentage of adult size that is present at birth) were compared across the four clades of mammals that are the best represented in our sample: artiodactyls, carnivorans, primates and rodents (Fig. 1a). All the species included in our study attain less than 21% of their adult body size at the time of birth, with minor differences across the major mammalian orders (Fig. 1b and Extended Data Table 1). However, other studies have reported higher body proportion values at birth in some groups that are not included in our study, such as some species of bat, which can reach almost 50% of their maternal body size at birth[28]. Among the phylogenetic groups we compared, body proportion at birth is significantly lower in carnivorans relative to artiodactyls (P = 0.011) and primates (P = 0.003), with no other significant differences observed.

In contrast, ranges of variation for brain proportion at birth are very wide. Within artiodactyls and primates, most species achieve between 25% and 75% of their adult brain size when they are born, with mean values of 40–50% (Extended Data Table 1). Humans show the lowest value within primates, with slightly less than 25% of adult brain size attained at birth. Carnivorans and rodents tend to show lower brain proportions at birth, with mean values around 25%, but their ranges of variation are also very wide, with maximum values reaching almost 75% in carnivorans and almost 60% in rodents (Fig. 1c and Extended Data Table 1). Primates display significantly higher brain proportions at birth than rodents (P < 0.001) and carnivorans (P < 0.001), but not artiodactyls (P = 0.943).

A similar pattern is observed when focusing on mammalian species with particularly large absolute adult brain sizes (Fig. 1d). These groups also show narrow ranges of variation for body proportion at birth, with the lowest overall values observed in hominids (great apes and humans) and elephants (Fig. 1e). These groups tend to show brain proportions at birth of around 40–50% (Fig. 1f and Extended Data Table 1), but humans and, especially, bears show particularly low values within their clades.

### Evolutionary rates

To assess the strength of selection over each branch of the mammalian phylogeny, we calculated the amount of change accumulated over each branch with respect to a neutral expectation. We call these values evolutionary rates because they are indicative of how fast individual branches have evolved, although they are not rates in the strict sense, but rather a ratio of the observed amount of change versus the expected amount of change at each branch (see Methods for detailed explanations on how these values have been calculated). Humans show the highest evolutionary rate towards a smaller brain proportion at birth (which we refer to as increased brain size altriciality) across all mammals (Fig. 2a,f, Extended Data Fig. 1 and Extended Data Table 2), with the same result obtained when brain size altriciality is measured as residuals from a phylogenetic generalized least squares (PGLS) regression line between neonatal and adult brain size (Extended Data Fig. 2). Although there are other mammalian species that have lower brain proportions at birth than humans, they have evolved their high levels of brain size altriciality within clades where other closely related species are similarly altricial. Humans, however, show a high level of brain size altriciality within the evolutionary context of primates, which are generally more precocial.

With respect to body proportion at birth, the species that shows the highest rate to increase body size altriciality (that is, to decrease body proportion at birth) is the orca (*Orcinus orca*), followed by the striped dolphin (*Stenella coeruleoalba*; Fig. 2b and Extended Data Table 2). Notably, the fast increase in brain size altriciality observed in humans is not a secondary result of an increase in body size altriciality, as humans show a moderate increase in body proportion at birth with respect to the last common ancestor of *Homo* and *Pan* (rate = 1.41; Fig. 2f). Indeed, humans have a higher body proportion at birth with

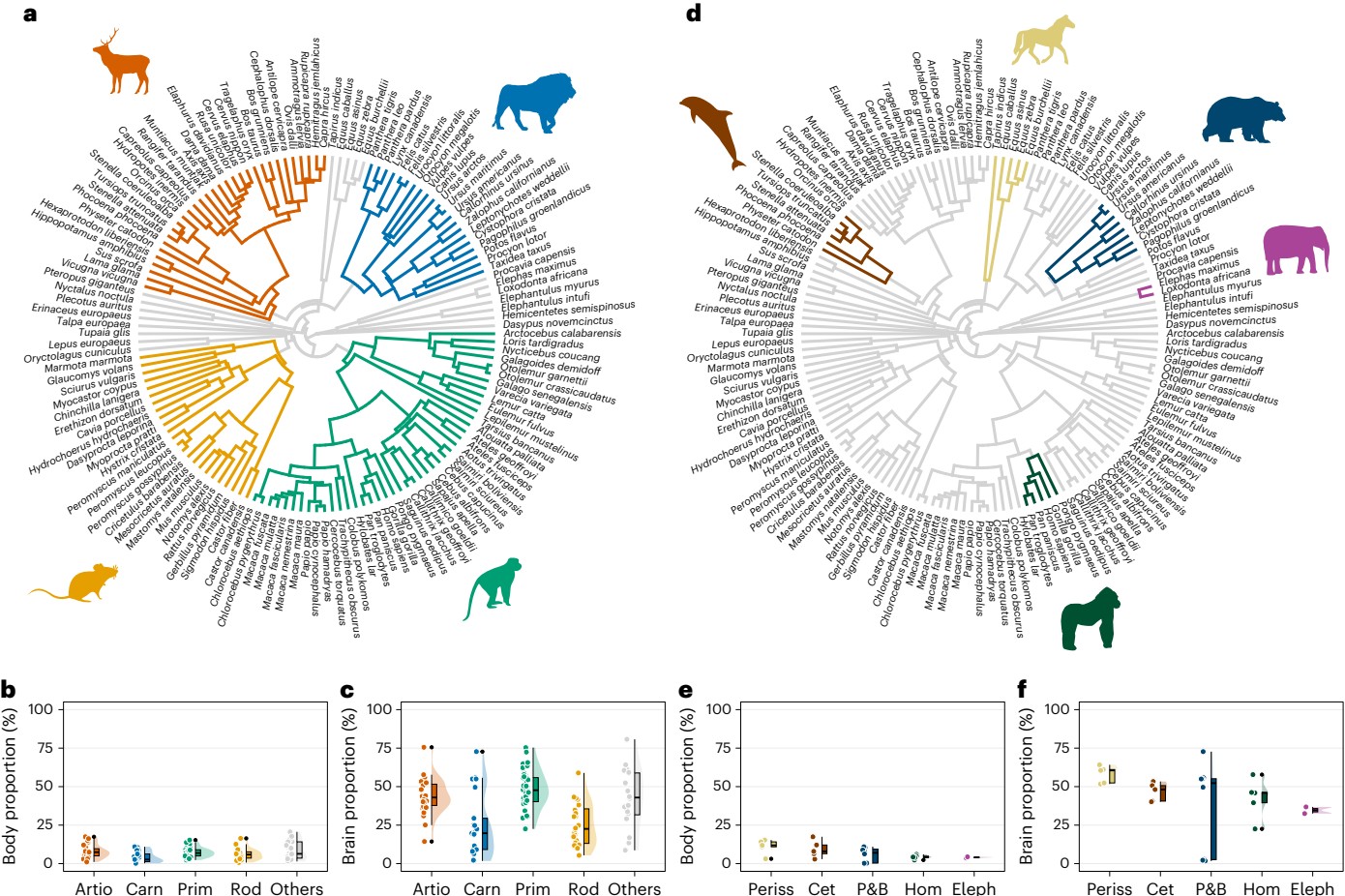

**Fig. 1 | Phylogeny and brain and body proportion values. a**, Phylogeny highlighting the four best-represented orders of mammals studied in the different analyses (artiodactyls in dark orange, carnivorans in blue, primates in green, rodents in light orange, others in grey). **b**, Comparison of body proportions at birth (percentage of maternal body size at birth) across the four orders of mammals. **c**, Comparison of brain proportions at birth (percentage of adult brain size at birth) across the four orders of mammals. **d**, Mammalian phylogeny highlighting the clades whose species have a particularly large absolute adult brain size (perissodactyls in yellow, cetaceans in maroon, pinnipeds and bears in dark blue, hominids in dark green, elephants in purple). **e**, Comparison of body proportions at birth across the five groups of mammals

with large brain sizes. **f**, Comparison of brain proportions at birth across the five groups of mammals with large brain sizes. Artio, artiodactyls (*n* = 26 species); Carn, carnivorans (*n* = 21 species); Prim, primates (*n* = 44 species); Rod, rodents (*n* = 24 species); Periss, perissodactyls (*n* = 5 species); Cet, cetaceans (*n* = 6 species); P&B, pinnipeds and bears (*n* = 8 species); Hom, hominids (*n* = 5 species); Eleph, elephants (*n* = 2 species). Raincloud plots show individual datapoints, probability density distributions and summary statistics in the box plots (median as the thick horizontal line, interquartile range within the box, minimum and maximum as the lower and upper whiskers, and outliers as black circles). Silhouettes are all from https://www.phylopic.org and are not to scale.

respect to maternal body size than all the other great apes (6.03% in humans versus 3.92% in chimpanzees, 4.83% in bonobos, 2.32% in gorillas and 4.25% in orangutans, according to our data). When considering our complete mammalian sample, evolutionary rates for brain and body proportions at birth are not significantly correlated (*r* = 0.007, *P* = 0.903; Fig. 3a), although they show a moderate correlation when outlier rates are removed (*r* = 0.272, *P* < 0.001; Fig. 3c). Within primates, these rates are not significantly correlated when outlier values are included in analyses (*r* = 0.001, *P* = 0.994; Fig. 3b), but they are positively correlated when outliers are excluded (*r* = 0.298, *P* = 0.006; Fig. 3d).

Although humans show a high rate to increase absolute adult brain size, this rate is not the highest observed across all mammals (Fig. 2c,f and Extended Data Table 2). The highest rate to increase adult brain size is observed in the branch leading to both species of elephants, and the second highest rate in the branch leading to fereuungulates (carnivorans, artiodactyls and perissodactyls). Evolutionary rates for brain proportion at birth and for absolute adult brain size have a negative borderline significant correlation, such that species that tend to

show high rates to increase their absolute adult brain size also tend to show high rates to decrease their brain proportion at birth (*r* = −0.117, *P* = 0.051; Fig. 3a), which is not surprising because brain proportion values also reflect changes in adult brain size. Across the complete mammalian sample, this negative correlation is also borderline significant when outlier rates are removed (*r* = −0.119, *P* = 0.053; Fig. 3c). This negative correlation is particularly strong within primates (*r* = −0.481, *P* < 0.001; Fig. 3b), although the strength of the correlation decreases when outlier rates, such as the human rate, are excluded (*r* = −0.241, *P* = 0.027; Fig. 3d). This negative correlation indicates that primate species showing high rates to increase adult brain size also tend to show high rates to decrease brain proportion at birth (in other words, they tend to increase brain size altriciality), a pattern that is not observed outside primates.

Humans have an evolutionary rate for absolute neonatal brain size of 1.59, which is indicative of a quasi-neutral increase with respect to the value that is inferred for the last common ancestor of chimpanzees and humans (Fig. 2d). As for gestation length, humans show a rate of 1.66 to increase the length of the gestation period with respect to the

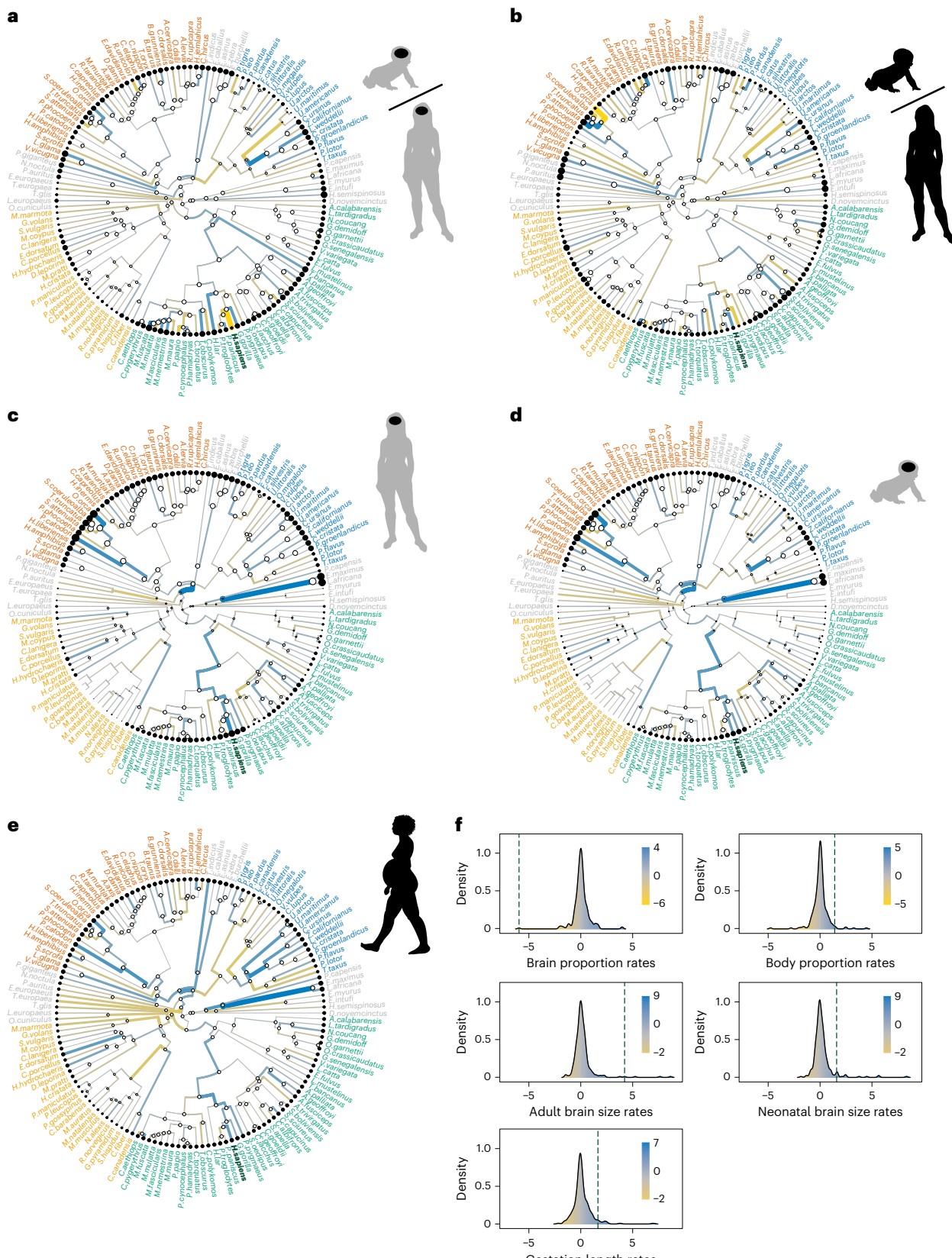

**Fig. 2 | Branch-specific evolutionary rates across the mammalian phylogeny.**
**a**, Brain proportion at birth. **b**, Body proportion at birth. **c**, Adult brain size.
**d**, Neonatal brain size. **e**, Gestation length. **f**, Comparison of the distribution
of evolutionary rates for each trait across the mammalian phylogeny with the
human rate (dark green dashed line). The layout of the plots in **f** is the same as in
the general figure. For branch colours, yellow indicates fast rates to decrease the
value of the trait under study, blue indicates high rates to increase the value of the
traits and grey indicates low rates. Nodes are represented in white and tips
are represented in black, with tip/node size proportional to the trait value
within each phylogeny. Species names are colour-coded according to their order
as in Fig. 1a, with humans highlighted in dark green. Silhouettes are all from
https://www.phylopic.org.

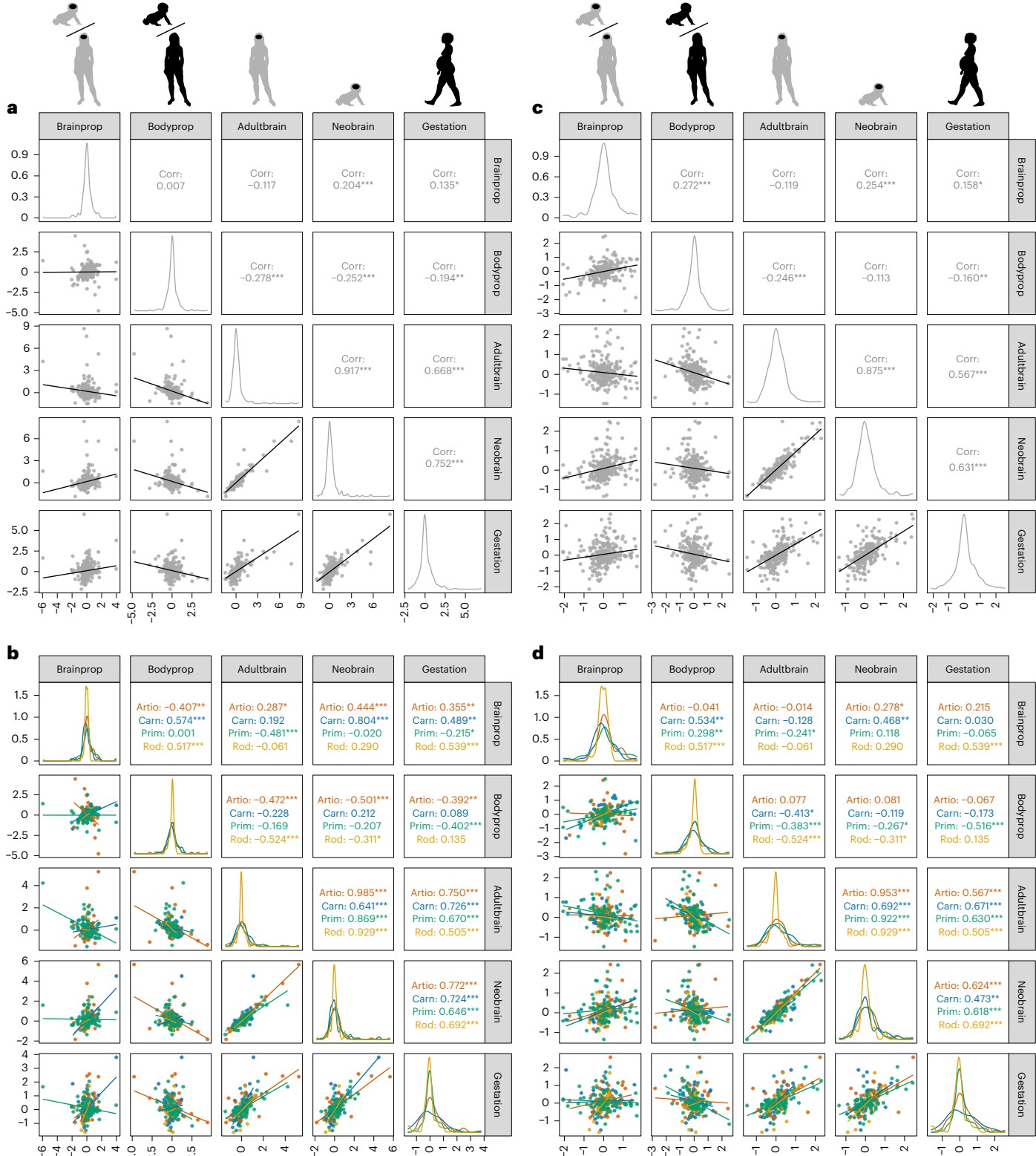

**Fig. 3 | Correlations between evolutionary rates.** Correlations between rates for brain proportion at birth (Brainprop), body proportion at birth (Bodyprop), absolute adult brain size (Adultbrain), absolute neonatal brain size (Neobrain) and gestation length (Gestation). **a**, All mammalian species are considered together. **b**, Correlations are measured within each of the four best-represented orders. **c**, All mammalian species are considered together, but outlier rate values (rate values greater than 3 or lower than −3) have been excluded from correlation analyses. **d**, Correlations obtained within each of the four best-represented orders after excluding outlier rate values. Asterisks indicate significant correlations at $P < 0.05$ (*), $P < 0.01$ (**) or $P < 0.001$ (***). Silhouettes are all from https://www.phylopic.org.

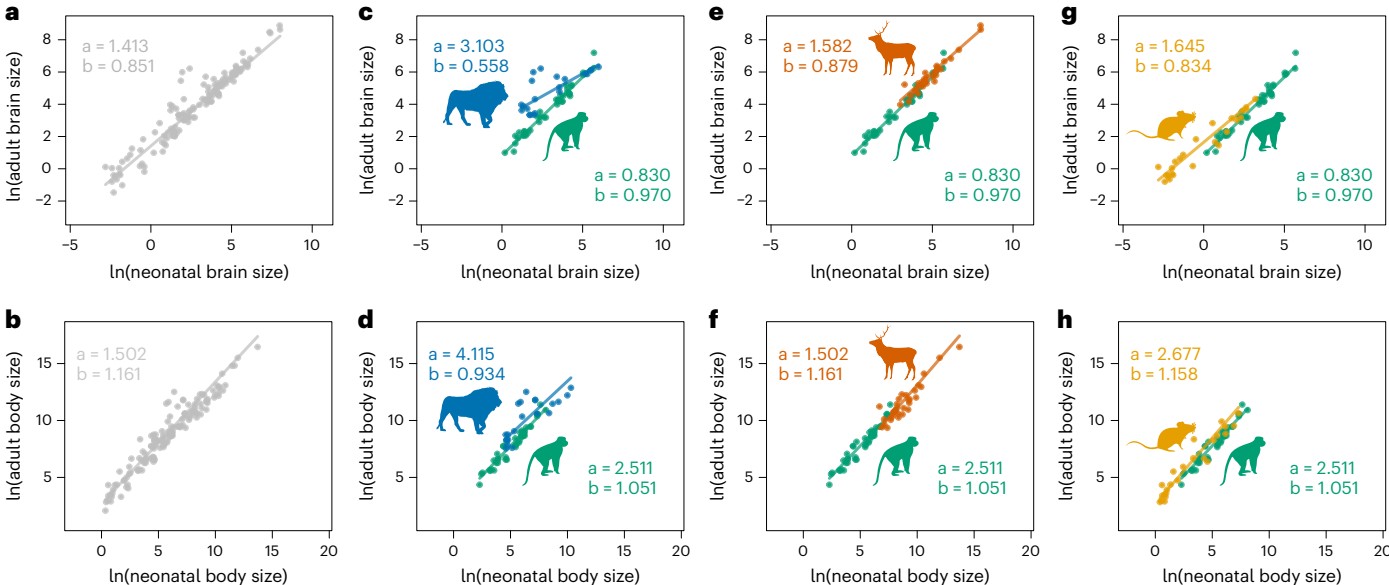

**Fig. 4 | PGLS regressions between neonatal and adult brain and body size.**
PGLS regressions between neonatal and adult brain size (top) and neonatal and adult body size (bottom) with comparisons between primates and the other three best-represented orders. Intercepts (a) and slopes (b) are indicated for each order, and they are compared based on pANCOVA analyses. **a**, Adult to neonatal brain size in the complete mammalian sample. **b**, Adult to neonatal body size in the complete mammalian sample. **c**, Adult to neonatal brain size in primates versus carnivorans ($F = 7.765$, $P < 0.001$). **d**, Adult to neonatal body size in

primates versus carnivorans ($F = 0.639$, $P = 0.531$). **e**, Adult to neonatal brain size in primates versus artiodactyls ($F = 0.911$, $P = 0.407$). **f**, Adult to neonatal body size in primates versus artiodactyls ($F = 0.341$, $P = 0.712$). **g**, Adult to neonatal brain size in primates versus rodents ($F = 1.573$, $P = 0.215$). **h**, Adult to neonatal body size in primates versus rodents ($F = 0.612$, $P = 0.545$). Mammalian orders are represented using the same colors as in Fig. 1, with all species represented in grey in the left-most panels. Silhouettes are all from https://www.phylopic.org.

last common ancestor of chimpanzees and humans, also indicating a neutrally evolving slight increase in gestation length with respect to the *Homo–Pan* ancestral value (Fig. 2e,f). Evolutionary rates for brain proportion at birth show a moderate positive correlation with rates for neonatal brain size and gestation length across the complete mammalian sample (Fig. 3a,c), which is not observed within primates (Fig. 3b,d).

### Scaling relationship between neonatal and adult brain and body size

The relationship between neonatal and adult brain and body size was also explored using PGLS regressions. Neonatal and adult body size scale with a slope similar to 1 in all mammalian orders, with generalized phylogenetic analysis of covariance (pANCOVA) procedures showing no significant differences in slope or intercept between the four orders (Fig. 4). Concurring with previous analyses of primate data[29], neonatal and adult brain size scale with a slope that is close to 1 in primates, rodents and artiodactyls, but not in carnivorans (Fig. 4). Indeed, primates and carnivorans differ significantly in their intercept and slope ($P < 0.001$), with carnivorans showing a lower slope than all the other orders. Despite their clear differences in absolute brain size and in their ecological specializations, we do not find any differences in the scaling relationship between neonatal and adult brain size between cetaceans and other artiodactyls ($P = 0.816$), between pinnipeds and terrestrial carnivorans ($P = 0.693$), between hominids and other primates ($P = 0.156$), or between murids and other rodents ($P = 0.338$; Extended Data Fig. 3).

pANCOVA analyses, however, show that humans differ significantly from all the other primates in their scaling relationship between neonatal and adult brain size ($P < 0.001$), but not in their scaling relationship between neonatal and adult body size ($P = 0.192$; Fig. 5a,b). A more detailed analysis of the relationship between neonatal and adult brain size between humans and non-human primates indicates that the change in the scaling relationship in humans is driven by a higher

than expected adult brain size, rather than by a lower than expected neonatal brain size (Extended Data Table 3).

Regarding other apparent outliers with respect to their orders, bears differ significantly from all the other carnivorans in the slope and intercept of their scaling relationship between neonatal and adult brain size ($P = 0.014$), and of their scaling relationship between neonatal and adult body size ($P = 0.033$; Fig. 5c,d). *Sus scrofa* also differs significantly from all the other artiodactyls in its relationship between neonatal and adult brain size ($P = 0.050$), and in its relationship between neonatal and adult body size ($P = 0.039$; Fig. 5e,f). Although visually not a clear outlier, the golden hamster (*Mesocricetus auratus*) differs significantly from other rodents in its relationship between neonatal and adult brain size ($P = 0.005$), and it shows the same trend in the difference of its scaling relationship between neonatal and adult body size, although this is not significant ($P = 0.058$; Fig. 5g,h). These results indicate that, out of the outliers observed for each order, humans are the only ones that show a scaling relationship between neonatal and adult brain size that is significantly different from the scaling relationship observed for the rest of their order, and that is not associated with a similar difference in the scaling relationship between neonatal and adult body size.

### Timing of neurodevelopment

Next, we aimed to understand how the timing of neurodevelopmental processes relative to birth changed during hominin evolution. To do so, we relied on Workman et al.'s model of neural development[23], which showed that the progress of neural events across 18 mammalian species (which span the phylogenetic diversity of mammals) is highly conserved. We used this model to calculate the day after conception at which key neurodevelopmental events occurred in fossil hominins, with particular focus on whether they happened before or after birth[23]. This timing was calculated based on the inferred adult brain size and gestation length of each hominin species (Extended Data Table 4; see Methods for more details).

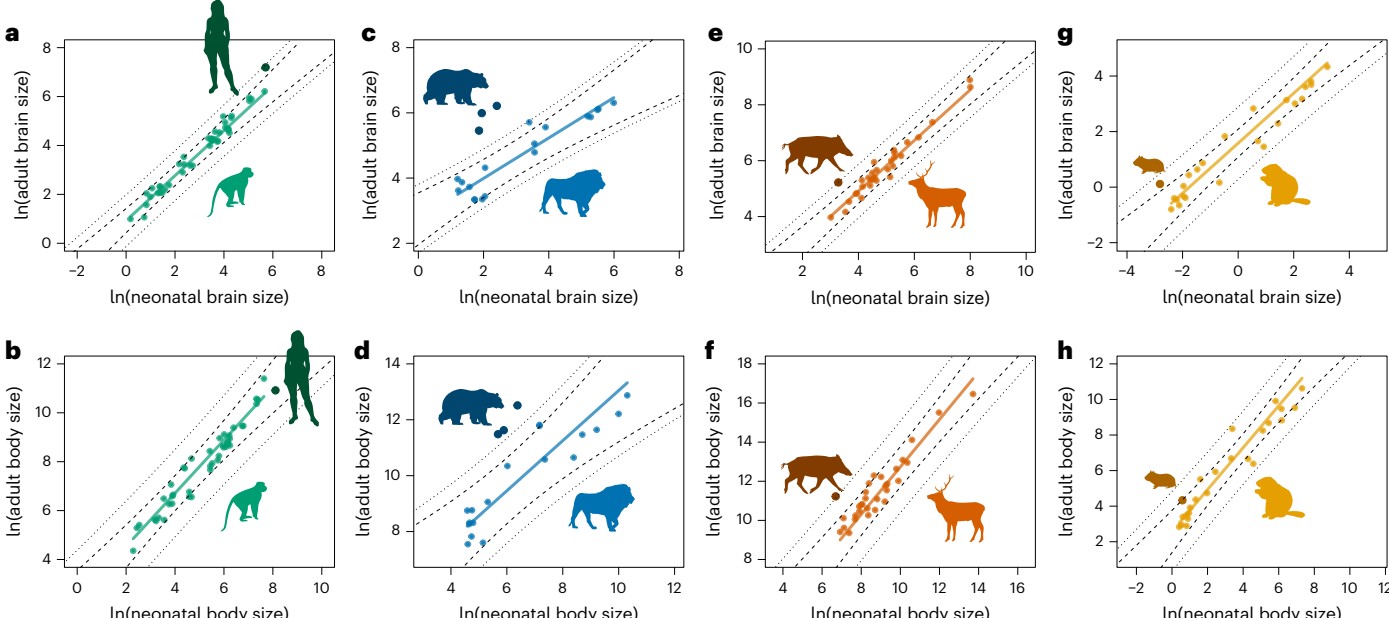

**Fig. 5 | Scaling relationships between neonatal and adult brain and body size in outlier species.** Comparison of the scaling relationship between neonatal and adult brain and body size for the four mammalian orders and their apparent outliers in their scaling relationship between neonatal and adult brain size, with significance assessed based on pANCOVA. **a**, Adult versus neonatal brain size in humans with respect to other primates ($F = 13.799$, $P < 0.001$). **b**, Adult versus neonatal body size in humans with respect to other primates ($F = 1.761$, $P = 0.192$). **c**, Adult versus neonatal brain size in bears with respect to other carnivorans ($F = 5.585$, $P = 0.014$). **d**, Adult versus neonatal body size in bears with respect to other carnivorans ($F = 4.178$, $P = 0.033$). **e**, Adult versus neonatal brain size in boars with respect to other artiodactyls ($F = 4.185$, $P = 0.050$). **f**, Adult versus neonatal body size in boars with respect to other artiodactyls ($F = 4.674$, $P = 0.039$). **g**, Adult versus neonatal brain size in golden hamsters with respect to other rodents ($F = 9.821$, $P = 0.005$). **h**, Adult versus neonatal body size in golden hamsters with respect to other rodents ($F = 4.018$, $P = 0.058$). Apparent outliers with respect to each order are represented with a darker shade of their order-specific colour. Confidence intervals (dashed) and prediction intervals (dotted) are plotted. Silhouettes are all from https://www.phylopic.org, with the exception of the golden hamster, which is self-generated.

Workman et al. calculated an 'event score' for each neurodevelopmental event that describes the order in which they occur[23]. This score ranges from 0 to 1, with 0 corresponding to the earliest occurring events (peak of neurogenesis of the cranial motor nuclei in the brainstem) and 1 corresponding to the latest occurring event (which corresponds to the end of the myelination of the middle cerebellar peduncle in their dataset)[23]. Using this model to study different hominin species shows that the duration of neurodevelopment is extended in modern humans and Neanderthals with respect to earlier hominins, and that modern humans and Neanderthals are born at an earlier neurodevelopmental stage not because gestation is shorter, but because neurodevelopment takes longer (Fig. 6a). This result is further confirmed when comparing the scale of neurodevelopment in humans with that corresponding to the great apes (Extended Data Fig. 4). Our results indicate that humans are born with an event score of 0.695, which compares with the event scores in the range of 0.728 to 0.756 shown by chimpanzees, bonobos, gorillas and orangutans at birth (Extended Data Fig. 4). If humans (present-day *Homo sapiens*) were born with the same event score as the other great apes (that is, at the same neurodevelopmental stage), this would correspond to an average gestation length of 321 days (10.7 months). Interestingly, the application of Workman et al.'s model[23] to the complete mammalian sample shows that there is a significant correlation between brain proportion at birth (percentage of adult brain size at birth) and neurodevelopmental stage at birth (measured as the event score with which species are born) across mammals ($r = 0.439$, $P < 0.001$), but there is no significant correlation within primates ($r = 0.289$, $P = 0.057$).

Focusing on fossil hominin species and on events related to brain development, there are only 13 out of 215 events (those with event scores ranging from 0.675 to 0.770) that change in their pre- or postnatal occurrence during hominin evolution, depending on species-specific combinations of brain size and gestation lengths (Fig. 6b and Extended Data Table 5). All events outside this range would have happened prenatally (when the event score is lower than 0.675) or postnatally (when the event score is higher than 0.770) in all hominin species. More specifically, events with lower event scores within the 0.675–0.770 range are the ones that were moved to the postnatal period as later hominins evolved larger brains. A closer examination of these events indicates that they are mostly related to the onset of myelination of different brain structures, including the anterior commissure, hippocampus, striatum and corpus callosum, among others (Fig. 6b and Extended Data Table 5).

## Discussion

Previous work has explored the relationship between neonatal brain size and several other factors, including maternal body size, maternal metabolic rate, overall maternal investment, gestation length, type of placentation and litter size in mammals[29–32]. These studies have tried to explain human altriciality relative to other primates as the result of certain evolutionary and developmental constraints. In contrast, our study has focused on brain development in an attempt to understand whether human altriciality has evolved as a selectively advantageous trait that may have increased brain plasticity and behavioural complexity[1,2].

Our results indicate that brain proportion at birth, which is generally considered a proxy for brain size altriciality, varies extensively within all mammalian orders included in our study. In addition, brain proportion at birth does not show a clear association with the neurodevelopmental stage at which species are born within primates. Therefore, although this percentage value is broadly used in the literature to discuss the evolution of human altriciality, our results indicate that

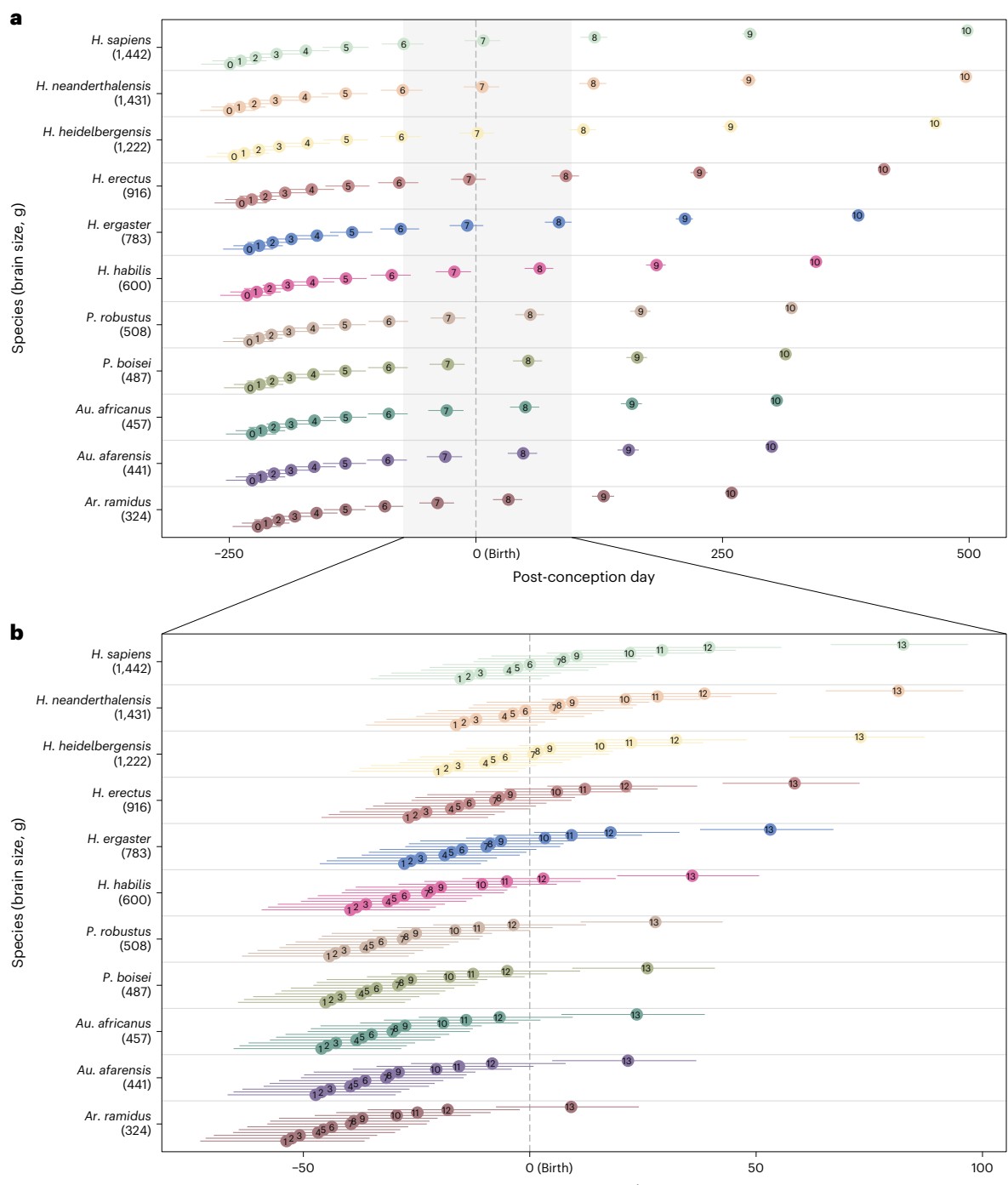

**Fig. 6 | Change in the timing of neurodevelopment across hominin evolution.**
**a**, Variation in the timing of neurodevelopmental events across Workman's complete scale (event score = 0 to event score = 1). Labels 0–10 in the plot represent Workman's event scores 0–1 (1 in the plot corresponds to an event score of 0.1 and so on). **b**, Variation in the timing of neurodevelopment of the events whose pre- or postnatal occurrence is inferred to change over hominin evolution, which corresponds to the shaded area in **a**. Only the events whose pre- or postnatal occurrence is inferred to change during hominin evolution are represented in **b**. Labels in the plot correspond to the following events and event scores (from ref. 23): 1, fornix myelination onset (0.675); 2, anterior commissure myelination onset (0.677); 3, lateral geniculate nucleus myelination onset (0.68); 4, cingulum myelination onset (0.687); 5, mammillothalamic tract myelination onset (0.689); 6, internal capsule myelination onset (0.692); 7, hippocampus myelination onset (0.699); 8, fasciculus retroflexus myelination onset (0.7); 9, stria terminalis myelination onset (0.703); 10, striatum myelination onset (0.715); 11, corpus callosum body myelination onset (0.722); 12, splenium myelination onset (0.732); 13, start of the plasticity/ocular dominance critical period (0.77). In both plots, the *x* axis represents the post-conception day at which events occur, with the plot centred on the day of birth (day 0), and the *y* axis represents hominin species in association with their adult brain size. The *y* axis in the plots is not to scale.

this proportion is not an accurate measure of the level of neurodevelopmental altriciality, particularly within primates. When altriciality is measured as this percentage value, humans appear to have evolved their high level of brain size altriciality at a very fast rate in the context of their more precocial primate relatives, showing the largest accumulated change with respect to a neutral expectation across our complete mammalian sample. Also, although the scaling relationship between neonatal and adult brain size appears to be highly conserved across

mammals (with the exception of carnivorans; Fig. 4), and particularly within primates, humans are the only species that departs significantly from the scaling relationship observed in their order (primates) in a way that is independent of the scaling relationship between neonatal and adult body size.

Given these results, we evaluated the likely drivers of the fast increase in brain size altriciality observed in humans. Our results indicate that the apparent high level of human brain size altriciality most strongly relates to a larger than expected adult brain size, rather than a smaller than expected neonatal brain size or a shortened gestation length. Indeed, humans show increased neonatal brain size and gestation length with respect to the last common ancestor shared with chimpanzees (Fig. 2). In taking body size scaling into account, published studies disagree on whether human gestation length is shorter than expected or not. Comparisons with other primates indicate that humans and the great apes have the gestation length that is expected for their body size[33], whereas other analyses indicate that human gestation length is shorter than expected in comparison with other primates, and that genes involved in parturition may show accelerated evolution in humans[34].

In our analysis, humans show a moderate rate of 1.66 to increase the length of the gestation period with respect to the last common ancestor shared with chimpanzees and bonobos. If the human gestation length was 7 months longer than it is, as expected if humans were to be born with a chimpanzee value of 40% of adult brain size at birth[35,36], this would require an evolutionary rate of 8.02 to increase the length of the gestation period (Extended Data Fig. 5). If the human gestation length was 21 months, which is often invoked as the gestation length that would be required for humans to be born at the same developmental stage of a chimpanzee[1], this would require a rate of 9.88 along the human branch. These rates would be the highest across all mammals and extreme outliers with respect to the rates measured within the primate clade for gestation lengths, which range from −1.18 to 2.05. Even our estimate of a gestation length of 10.7 months, which would be required for humans to be born at the neurodevelopmental stage of the other great apes according to Workman's model[23], would require an evolutionary rate of 3.5 along the human branch, which is within the range of variation of mammals, but outside the range of variation observed in primates. These comparisons show that, given the evolutionary context of the variation in gestation length across primates, humans could not have evolved the very long gestation periods that align with what would be expected for the brain size precociality observed in other primate species.

Previous work has shown that pre- and postnatal brain growth rates show a strongly conserved pattern across placental mammals, with brain size at birth simply capturing one particular time point within the dynamic process of neurodevelopment[37]. The timing of birth with respect to neurodevelopment, however, is known to be highly variable across clades, but also between closely related species[38], as also shown by our analyses. Previously published work on a small but diverse sample of mammalian species[37] shows that peak growth velocity happens postnatally in all altricial species and prenatally in all precocial species included in the sample, and also in humans. However, birth happens closer to peak growth velocity in humans than in other precocial species[37], which results in the accelerated brain growth observed in humans during the first year of life in comparison with chimpanzees[39,40] and other primates[41]. In spite of these differences in postnatal brain growth rates, comparisons between humans and chimpanzees[42] and other great apes[43] based on anatomy, behaviour and transcriptional profiles indicate that humans and great apes are in many ways similar from a developmental point of view during the first year of life, and that differences between species become more marked during later stages of maturation.

Still, humans are often described as developmentally delayed at birth in comparison with other primates. Comparisons of humans and chimpanzees indicate that gross motor control, including the ability to sit up, stand up, walk and climb, develops faster in chimpanzees[44], although other studies indicate that the timing of walking onset is accurately predicted by brain size in humans and other mammals[45]. Fine motor control and social behaviour seem to emerge at similar paces in chimpanzees and humans, particularly when differences in lifespan and age at first reproduction are taken into account[44]. Focusing on the early postnatal period (that is, the first 30 days after birth), behavioural studies of captive chimpanzees have shown that they differ significantly from humans in only 1 of the 25 traits assessed by the Neonatal Behavioural Assessment Scale, namely in muscle tone[46] (assessed traits are related to attention, arousal, motor ability and coping). These comparisons indicate that the assertion that humans are developmentally and behaviourally more altricial than other primates is primarily based on gross motor development, but both newborn and adult humans show substantially less motor strength than their great ape counterparts (also discussed in ref. 35). More comparable data with respect to other behavioural domains are needed to clarify the nature of differences during the early postnatal period between humans and other primates.

Confirming previous studies[23,38,47], our analyses show that the duration of neurodevelopment is longer in modern humans and Neanderthals with respect to earlier hominins with smaller brains (Fig. 6a). This effect is amplified towards the later occurring events and, therefore, is particularly marked during the postnatal period. In fact, this extended duration of neurodevelopment drives the apparent altriciality of humans to a much stronger degree than a change in the length of the gestation period, which remains fairly stable among hominids (great apes and humans)[12,33]. Our analyses indicate that approximately 6% of all the neurodevelopmental events studied by Workman et al.[23] were shifted to the postnatal period during hominin evolution (that is, they would have happened prenatally in early hominins, including *Ardipithecus ramidus* and australopiths, and postnatally in later hominins, including Neanderthals and modern humans; Fig. 6b). Those events are mostly related to the onset of the myelination of some brain regions (including the corpus callosum, hippocampus and striatum, among others), which adds to the existing evidence that there are changes in the development of white matter between chimpanzees and humans that evolved after their divergence[39,40]. The pre- or postnatal occurrence of some other neurodevelopmental events related to brain plasticity, such as those related to synaptogenesis and attainment of peak synaptic density, seem to be shared by both chimpanzees and humans[42,48] and, according to our estimates, by fossil hominins.

Together with observations that myelination is developmentally protracted in humans with respect to chimpanzees[49], our results indicate that evolved differences in the timing of myelination involve the complete period of postnatal development in humans, from birth to early adulthood. Because differences in brain plasticity between chimpanzees and humans are well described at the anatomical and molecular level[50–56], our results point to a particularly important role of myelination in driving human brain plasticity. Activity-dependent myelination is a mechanism of brain plasticity that can optimize the timing of information transmission through neural circuits and that can be increasingly important in large brains with complex networks, such as human brains, where conduction delays are substantial and the synchronous arrival of action potentials is critical for optimal network function[57].

Differences in human brain plasticity further develop later in the postnatal period. The existence of a critical window during the early infancy for the onset of a variety of cognitive functions and species-specific social behaviours is widely documented in humans[58] and other primates[59], and is also demonstrated by studies showing the effect of early rearing experience on brain structure in humans[60], chimpanzees[61] and rhesus monkeys[62]. In humans, some key developmental milestones that do not have a clear parallel in other primates are

attained during the first year of life, including those that are related to the emergence of language[63] and shared intentionality[64]. By two years of age, humans show significant differences with chimpanzees and bonobos in several aspects related to social cognition, which further develop afterwards[65]. The small proportion of neurodevelopment that has been moved from the prenatal to the postnatal period during human evolution seems to indicate that human specializations for neuroplasticity are more strongly linked to later stages of neurodevelopment, rather than to the early postnatal period. However, this small number of events may have had a large functional significance, and their effects can be increasingly pronounced as neurodevelopment progresses into later postnatal stages.

In summary, our results suggest that humans are not exceptionally altricial in comparison with other species[12], and they help us understand the aspects of early postnatal brain development that have changed during human evolution. While human brains are slightly less developed at birth than expected within their phylogenetic context, this is not because of a shortened gestation, but because of the longer extension of neurodevelopment, which is ultimately linked to increased adult brain size (that is, larger brains need more time to grow). Our results indicate that only a minor proportion of neurodevelopment was shifted from the prenatal to the postnatal period during human evolution, but additional data are required to elucidate whether this apparently minor shift may have had substantial functional effects. More specifically, the slight underdevelopment of human brains at birth is consistently associated with the postnatal occurrence of some developmental events related to myelination that happened prenatally in earlier hominins. Our results point to the interaction between myelination, environmental influences during postnatal development and brain plasticity as a fruitful avenue for future research to shed light on the evolution of human-specific behavioural traits.

## Methods

### Sample and phylogeny

We studied the relationship between neonatal and adult brain and body size in a broad sample of placental mammals whose position along the altricial–precocial spectrum spans all the diversity observed in this clade (Fig. 1a). These species differ widely in their absolute adult brain size from less than 1 g in some bats and rodents to more than 7,000 g in the sperm whale, and in their absolute adult body size from 10–20 g in some rodents and bats to 14,000 kg in the sperm whale[66]. A sample of 140 species (including 44 primate species, 24 rodent species, 21 carnivoran species, 26 artiodactyl species and 25 species from other mammalian orders) was used. Artiodactyla in our study includes both artiodactyls and cetaceans, a clade that is sometimes termed Cetartiodactyla[67]. Data on species' mean neonatal and adult brain and body size, and well as on gestation length and generation times, were compiled for all the species from different sources. Data on neonatal and adult brain and body size were obtained from refs. 68,69. For those values that were missing or looked too different from those reported in other publications, values were added or double-checked and amended as necessary using refs. 30,34,70 for brain size data and refs. 28,30,66,71 for body size data. Data on gestation lengths and generation times were obtained from refs. 66,71, with age at first reproduction used as a proxy for generation time. The species-specific values we obtained from the literature did not represent longitudinal data on neonatal and adult brain and body sizes from the same individuals. Consequently, like previous comparative research, the relationship between neonatal and adult brain and body size in our study is influenced by intraspecific variation. Adult body size values corresponded to maternal values, whereas neonatal brain and body size values, as well as adult brain size values, were normally not assigned to males or females in the source datasets.

Given that some of these datasets were published decades ago, species names were checked and amended as needed to make sure that they matched current taxonomic views as reflected in the employed phylogeny (see below). Species were not included when they had missing data for one or more variables. In a few cases, gestation lengths and generation time values corresponding to a given species were obtained from their closest sister species within the same genus. We used a recently estimated mammalian phylogeny[72], which was pruned to include only the species included in our dataset.

### Analysis of extant mammals

We first tested whether the best-represented orders within our dataset (Primates, Carnivora, Rodentia and Artiodactyla) differ in their brain and body proportion at birth, defined as the proportion of adult brain (or body) size that is represented by neonatal brain (or body) size. Significance was assessed based on pairwise Wilcoxon rank sum tests with Bonferroni correction. This comparison was carried out for the complete sample and for a selection of those species whose brain sizes are particularly large, which are grouped in five different clades at different taxonomic levels: cetaceans, hominids (great apes and humans), elephants, perissodactyls, and the clade formed by pinnipeds and bears. The mammalian phylogeny used in our study includes pinnipeds and bears as sister clades, with musteloids forming a sister group to both of them. Other mammalian phylogenies, however, consider musteloids as the sister group of pinnipeds, with bears as a sister group to the pinniped–musteloid clade[73]. The sample size of these groups ranged from two (elephants) to eight (pinnipeds–bears).

### Evolutionary rates

To infer the strength of selection over each branch of the mammalian phylogeny, we calculated the amount of change accumulated over each branch relative to the expected amount of change per branch, which depends on branch length (that is, longer branches are expected to accumulate more change than shorter branches). As a first step, we calculated ancestral values at each node of the phylogeny. To do so, we used a variable rates approach implemented in the software BayesTraits V3[74,75]. The variable rates model was used with the default priors, and it was run for 10 million iterations with a 20% burn-in period. BayesTraits' variable rates model detects shifts from an underlying homogenous Brownian motion model of evolution in a phylogeny without prior knowledge of where those shifts have occurred[74,76]. This model finds a set of branch length scalars that optimize the fit of the data to a homogenous Brownian motion model of evolution, which results into a rescaled tree where each branch has been stretched or compressed to conform to a Brownian motion process[76]. Stretched branches represent fast evolutionary change, whereas compressed branches represent slow evolutionary change for a given trait.

The obtained rescaled trees were used to calculate the most likely ancestral value at each node of the mammalian phylogeny using the package ape[77] in R. The amount of change accumulated over each branch was then calculated as the difference between each descendant and ancestral value, and it was compared with the amount of change that each branch would have accumulated had they evolved at the same rate[26]. This expected amount of change was calculated for each branch of the phylogeny as a constant tree-wide per-generation variance parameter (estimated from the values observed in our dataset for each trait) multiplied by the square root of branch lengths after transforming each branch of the phylogeny to generations[78]. This transformation was attained using the generation time typical of each species and the reconstructed ancestral generation times[26]. A ratio was then calculated between the observed and expected amount of change per branch. This ratio has an absolute value of 1 when the observed and expected amounts of change are the same for a given branch. An absolute value higher than 1 indicates that branches accumulated more change (and, therefore, evolved faster) than expected, and an absolute ratio between 0 and 1 indicates that branches accumulated less change (and, therefore, evolved slower) than expected[26], with a positive or negative sign indicating trait increase or decrease with respect to the ancestral value.

A value close to 0 indicates that a given branch has not changed from its ancestral to its descendant value, then indicating evolutionary stasis.

Our approach results in distributions of ratios of observed versus expected change that are directly comparable across different traits. These values can be understood as evolutionary rates because they indicate how fast evolutionary change has accumulated, but they are not rates in the strict sense, as they do not measure change per unit of time. Rather, these ratios measure the amount of change accumulated along each branch of the mammalian phylogeny with respect to a neutral expectation, and they indicate both the strength and the directionality of evolutionary change by comparing each descendent value with its ancestral value. This approach is conceptually similar to that we have used in previous publications[26,79], but it relies on BayesTraits' variable rates approach to calculate ancestral values. Phylogenetic analyses and visualization of results also relied on the packages phytools[80], geiger[81], ggplot2[82] and smplot2[83].

Apart from being biologically reductionist and not fully reflective of the complex distinction between altriciality and precociality, brain proportion at birth as a proxy for brain size altriciality is also problematic from a purely quantitative point of view. Percentage values often show distributional issues that may violate the assumptions of subsequent statistical analyses, they increase the measurement error and they may introduce spurious correlations with other variables, among other statistical issues[84–86]. However, as mentioned in the introduction, we deemed it important to explore the variation of brain and body proportions at birth across mammals because of the historical importance of this value in discussions about human altriciality. We compared the evolutionary rates obtained for brain and body proportion at birth with those obtained when measuring brain and body size altriciality as the residuals obtained from a phylogenetic regression between neonatal and adult size. The evolutionary rates obtained based on brain and body proportions at birth are highly correlated with those based on PGLS regression residuals (Extended Data Fig. 2).

Brain and body proportions at birth were arcsine squared root transformed, and absolute adult brain size, absolute neonatal brain size and gestation length (in days) were log-transformed for the measurement of evolutionary rates. Pearson's correlations between evolutionary rates for brain proportion at birth, body proportion at birth, absolute brain size in adults, absolute brain size in neonates and gestation length were measured across the complete mammalian sample and within each of the four best-represented orders. Because correlations can be influenced by extreme rate values, these correlations were measured twice, first including all the rates and later after removing outliers, corresponding to rates lower than −3 or higher than 3. Before measuring these correlations, we tested whether unsigned rates obtained for the five variables are significantly correlated with branch lengths (BL). Two of these correlations show significant $P$ values, but we did not find a consistent negative correlation between BL (measured as millions of years or as generations) and evolutionary rates, which indicates that high rates are not preferentially found in short branches (brain proportion at birth versus BL (Myr): $r = -0.022$, $P = 0.719$; brain proportion at birth versus BL (generations): $r = -0.096$, $P = 0.111$; body proportion at birth versus BL (Myr): $r = -0.040$, $P = 0.505$; body proportion at birth versus BL (generations): $r = -0.133$, $P = 0.026$; absolute adult brain size versus BL (Myr): $r = 0.093$, $P = 0.120$; absolute adult brain size versus BL (generations): $r = -0.099$, $P = 0.099$; neonatal brain size versus BL (Myr): $r = 0.108$, $P = 0.071$; neonatal brain size versus BL (generations): $r = -0.098$, $P = 0.102$; gestation length versus BL (Myr): $r = 0.240$, $P < 0.001$; gestation length versus BL (generations): $r = -0.056$, $P = 0.349$).

### Scaling relationships between neonatal and adult size
To further explore the evolution of developmental patterns across mammals, the scaling relationship between neonatal and adult brain and body size was compared across the four best-represented orders

using PGLS regression analysis. This comparison used a generalized pANCOVA approach[87] implemented in the R package evomap[88] to test for differences in intercepts and slopes among those orders. This approach allows testing whether individual species, or groups of species, deviate significantly from the expected scaling relationships observed in the other species in their clade. The scaling relationships between these variables were also tested in the species or clades that looked like apparent outliers with respect to the order they belong to, that is, humans with respect to other primates, bears (clade formed by *Ursus arctos*, *U. maritimus* and *U. americanus*) with respect to other carnivorans, boars (*Sus scrofa*) with respect to other artiodactyls and golden hamsters (*Mesocricetus auratus*) with respect to other rodents. For humans, we further assessed whether changes in the scaling relationship between neonatal and adult brain and body size were primarily driven by neonatal or adult values. This was attained by comparing the human neonatal and adult values with the *Homo–Pan* estimated ancestral values relative to the scaling coefficient of the non-human primate scaling relationship as described in ref. 89. Within each order, we also compared the scaling relationship between neonatal and adult brain and body size values between subclades that differ substantially in adult brain size and/or ecological specializations (hominids versus other primates, pinnipeds versus terrestrial carnivorans, cetaceans versus other artiodactyls and murids versus other rodents). Following the temporal sequence of ontogenetic development, our regression analyses consider neonatal values as the independent variable and adult values as the dependent variable.

### Timing of neurodevelopment
Workman et al.'s model was used to infer the pre- or postnatal occurrence of neurodevelopmental events in fossil hominin species[23] (we use the term hominin to refer to fossil species that are more closely related to humans than to chimpanzees, whereas we use the term hominid to refer to the clade formed by humans and the great apes). The model estimates the day after conception at which a particular neurodevelopmental event would happen in a given species as:

$$Y = \text{intercept} + \text{slope} \times \text{eventscale} + (\text{interaction term}) \quad (1)$$

where $Y$ is the log-transformed day after conception at which a given event happens and 'eventscale' is the event score calculated by Workman et al. for each of the 271 neurodevelopmental events included in their study (which can be found in Table 1 of ref. 23). Out of those 271 events, we focused on the 215 events that are specifically related to brain development, which are allocated to the brainstem, cerebellum, limbic system, thalamus, striatum and cortex. We excluded the events that are described as increases in brain size, as variation in the sensory periphery and retina, and those classified as behavioural responses of the whole organism. While the list of neurodevelopmental events studied by Workman et al. is necessarily limited and cannot accurately reflect the whole complexity of neurodevelopment[23], it is still the most complete compilation that is available for comparison across species. For neurogenetic events happening in the cortex of non-glire mammals (which include primates and, therefore, hominins), an interaction term of 0.263 is added, although this is not relevant to our study because all neurogenetic events are early occurring events that happen prenatally in all primates[23], so we did not study them in detail. The intercept and slope in equation (1) are species-specific values that are calculated for each species as follows, according to the empirical relationships inferred by Workman et al.[23]:

$$\text{Species intercept} = 1.241 + 0.368 \times \log(\text{gestation length}) \quad (2)$$

$$\text{Species slope} = 1.474 + 0.257 \times \log(\text{adult brain mass}) \quad (3)$$

where 'gestation length' is the species-specific average gestation length measured in days, and 'adult brain mass' is the species-specific average

adult brain weight measured in grams. The pre- or postnatal occurrence of each neurodevelopmental event was calculated simply by subtracting the species-specific gestation length from the post-conception day at which each event is estimated to happen:

$$\text{Pre- or postnatal day} = \exp(Y) - \text{gestation length} \qquad (4)$$

A positive value indicates a postnatal occurrence, whereas a negative value indicates a prenatal occurrence.

Adult brain weight can be reliably estimated for most hominin species based on endocranial volume. Species-specific endocranial volumes were obtained from refs. 18,27, and they were transformed to brain masses in grams following ref. 90 (Extended Data Table 4). Following ref. 27, species-specific gestation lengths in hominins were calculated as the neonatal body mass typical of each species (obtained from ref. 18) divided by the prenatal growth rate estimated in ref. 27. This approach consistently indicates that earlier hominin species had shorter gestation lengths than later hominin species[27]. However, gestation length values obtained using this approach show unrealistically extreme values that range from a mean gestation length of 168 days (5.6 months) in *Ar. ramidus* to 313 days (10.4 months) in Neanderthals (Extended Data Table 4), which are far from the range of variation observed in all the great apes and humans (from 227 days in chimpanzees to 275 days in humans). Therefore, the obtained gestation lengths were rescaled to the interval 245–275 days to obtain an adjusted gestation length for each species. The lower bound of this interval corresponds to the gestation length calculated for the last common ancestor of chimpanzees and humans based on the variable rates approach described above (245 days), and the upper bound corresponds to the value observed in modern humans (275 days)[66]. While, based on their larger adult brain size, it is possible that Neanderthals and fossil modern humans had a longer average gestation than present-day modern humans, their gestation lengths are unlikely to have differed radically from that observed in present-day humans.

An error range for the gestation length value was calculated by using a minimum and maximum neonatal body mass estimate based on a human model and an ape model from ref. 18, respectively. A minimum and maximum gestation length was calculated using the minimum and maximum neonatal body masses. The percentage values with respect to the mean gestation length represented by the minimum and maximum values were subtracted or added to the adjusted mean gestation length to obtain an adjusted minimum and adjusted maximum gestation length. The error associated with other variables involved in the analysis of the timing of neurodevelopment, such as adult brain size and Workman's event score, was not included in the analysis, as those values are more directly based on empirical data. While this does not imply that these variables are free of error or variation, it does mean that their values are known with more accuracy than that of the gestation length of fossil hominin species.

Workman's equations were also used to calculate the event score at which each mammalian species is born by making *Y* equal to the day of birth of each species (that is, to their log-transformed gestation length) and solving 'eventscale' in equation (1) above. The intercept and slope in equation (1) were calculated from equations (2) and (3) using the average brain size and gestation length corresponding to each species. The gestation length that would be expected in humans if they were born at the same neurodevelopmental stage as the great apes was also calculated using equation (1) by making 'eventscale' equal to the event score at birth of each great ape species and solving *Y*.

## Reporting summary

Further information on research design is available in the Nature Portfolio Reporting Summary linked to this article.

## Data availability

Datasets used in this study are available at https://doi.org/10.6084/m9.figshare.22242724.

## Code availability

The scripts used to carry out analyses are available at https://doi.org/10.6084/m9.figshare.22242724.

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

## Acknowledgements

We thank C. Charvet, A. Goswami, T. Monson and J. DeSilva for clarifications on different aspects of this study. A.G.-R. is grateful to G. and V. Gómez-Sánchez for their practical lessons on human development. C.C.S. is supported by the National Science Foundation (EF-2021785, DRL-2219759) and the National Institutes of Health (HG011641).

## Author contributions

A.G.-R. and C.C.S. conceived the study. A.G.-R. designed research. A.G.-R. and C.N. compiled data. A.G.-R., C.N. and J.B.S. analysed data. All the authors interpreted results. A.G.-R. and C.C.S. wrote the manuscript with contributions from the other authors.

## Competing interests

The authors declare no competing interests.

## Additional information

**Extended data** is available for this paper at https://doi.org/10.1038/s41559-023-02253-z.

**Correspondence and requests for materials** should be addressed to Aida Gómez-Robles.

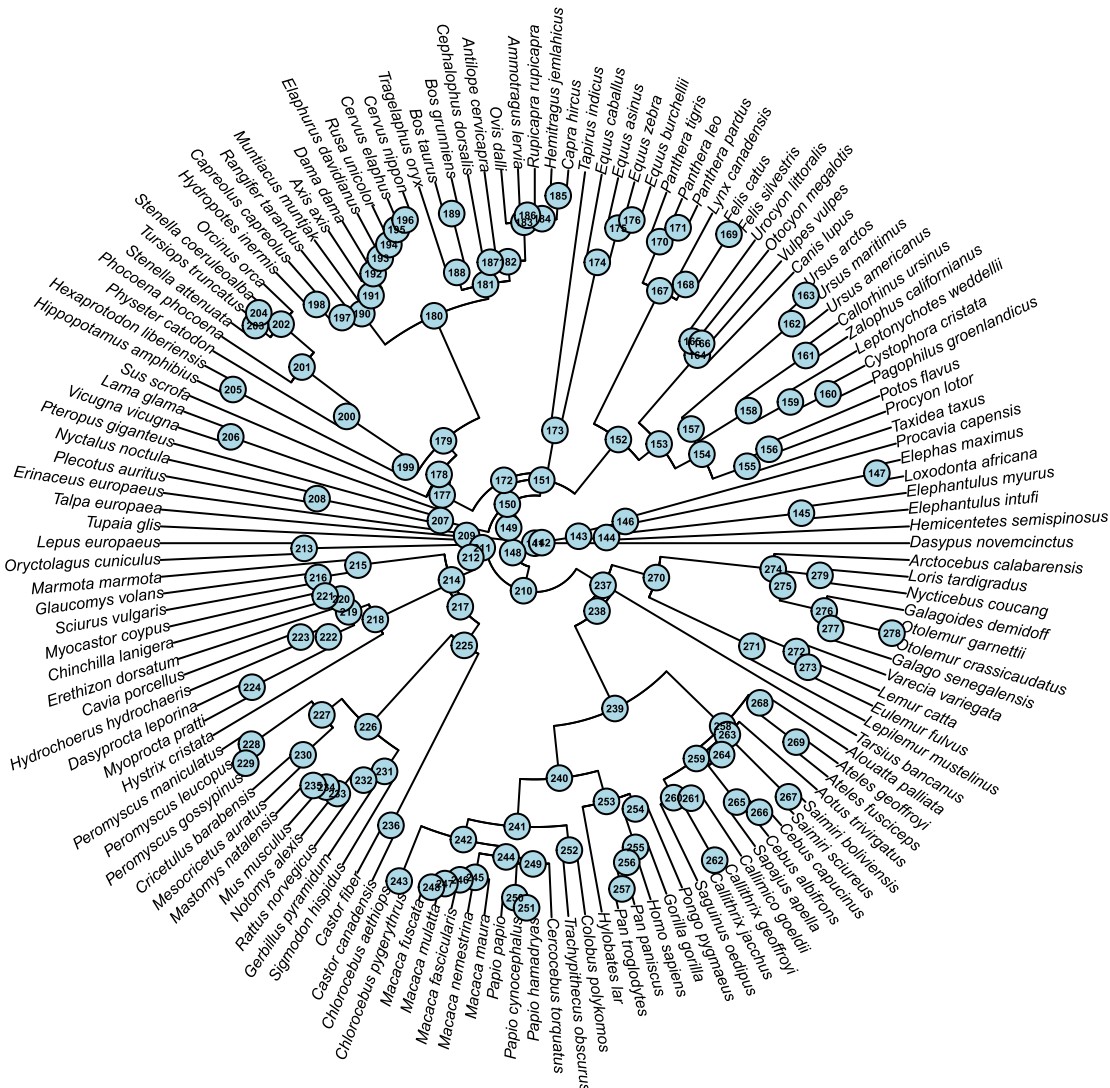

**Extended Data Fig. 1 | Phylogeny with node numbers.** Node numbers in this figure match those listed in Extended Data Table 2.

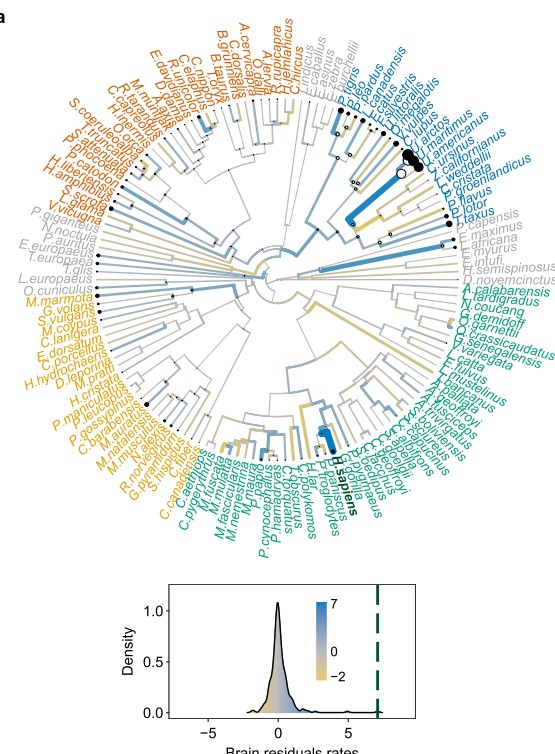

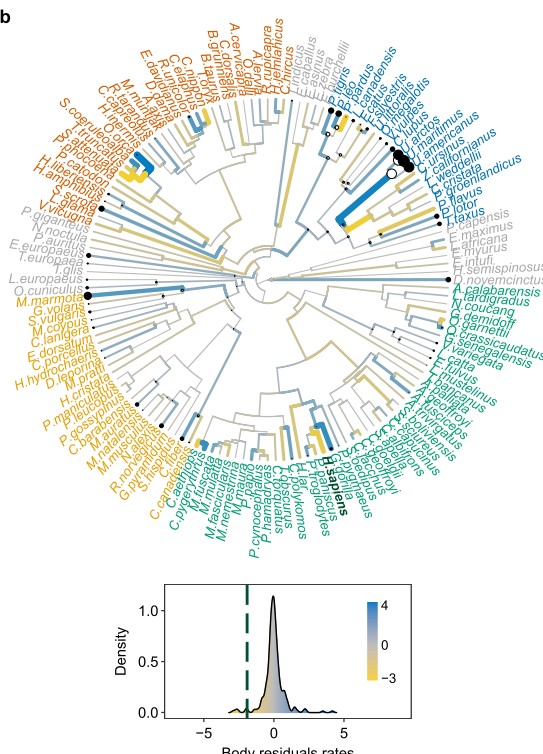

**Extended Data Fig. 2 | Evolutionary rates obtained when quantifying altriciality-precociality as the residuals from a PGLS regression between neonatal and adult brain and body size. a** Evolutionary rates for neonatal-adult brain residuals plotted on the phylogeny (top) and distribution of rates compared with the human evolutionary rate (dark green line, bottom). **b** As **a**, but for body residuals. Both sets of rates are strongly correlated with those obtained when calculating rates for brain and body proportions at birth as percentage values of neonatal to adult size, shown Fig. 2a and b, respectively (brain: $r = -0.833$, $P < 0.001$; body: $r = -0.934$, $P < 0.001$). Increased altriciality is indicated by a decrease in brain and body proportion values, but by an increase in brain and body residuals, hence the negative correlations.

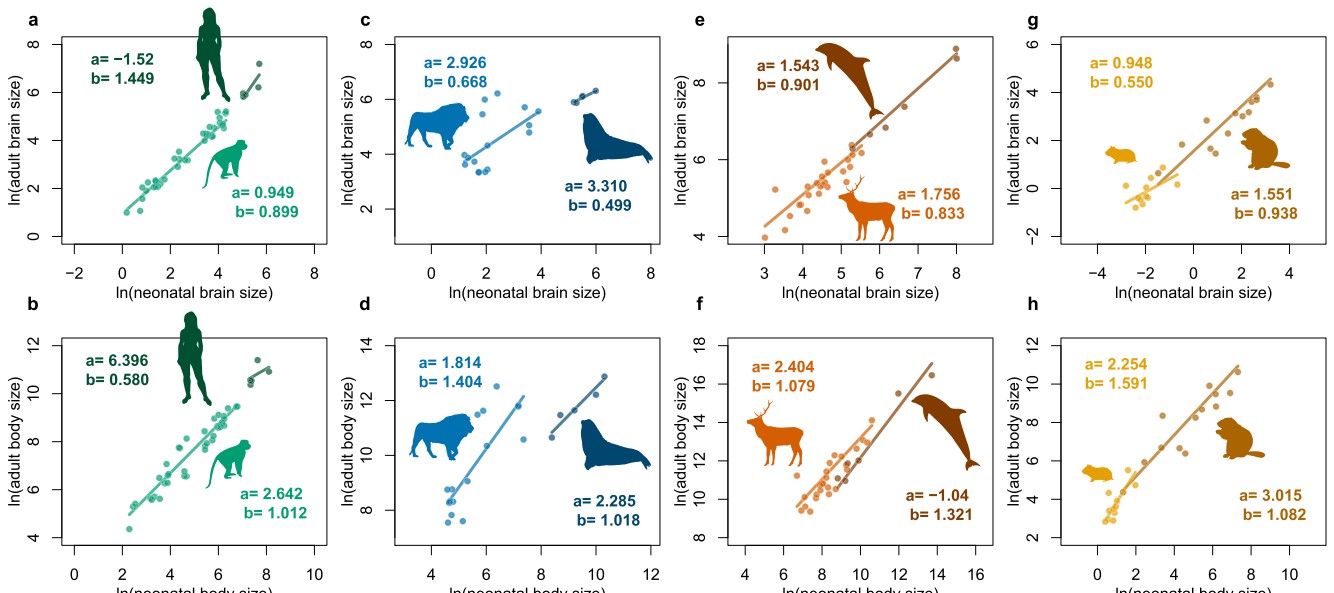

**Extended Data Fig. 3 | Scaling relationship between neonatal and adult brain and body size in mammalian subclades.** pANCOVA-based comparisons of the scaling relationship between neonatal and adult brain and body size in clades that differ substantially in adult brain size and/or ecological specializations within each order. **a** Adult versus neonatal brain size in hominids with respect to other primates ($F = 1.944$, $P = 0.156$). **b** Adult versus neonatal body size in hominids with respect to other primates ($F = 1.062$, $P = 0.355$). **c** Adult versus neonatal brain size in pinnipeds with respect to other carnivorans ($F = 0.375$, $P = 0.693$). **d** Adult versus neonatal body size in pinnipeds with respect to other carnivorans ($F = 5.580$, $P = 0.030$ for differences in slope). **e** Adult versus neonatal brain size in cetaceans with respect to other artiodactyls ($F = 0.205$, $P = 0.816$). **f** Adult versus neonatal body size in cetaceans with respect to other artiodactyls ($F = 1.304$, $P = 0.288$). **g** Adult versus neonatal brain size in murids with respect to other rodents ($F = 1.145$, $P = 0.338$). **h** Adult versus neonatal body size in murids with respect to other rodents ($F = 0.806$, $P = 0.460$). Groups with a larger adult brain size within each order are represented with a darker shade of their order-specific color. Silhouettes are from https://www.phylopic.org.

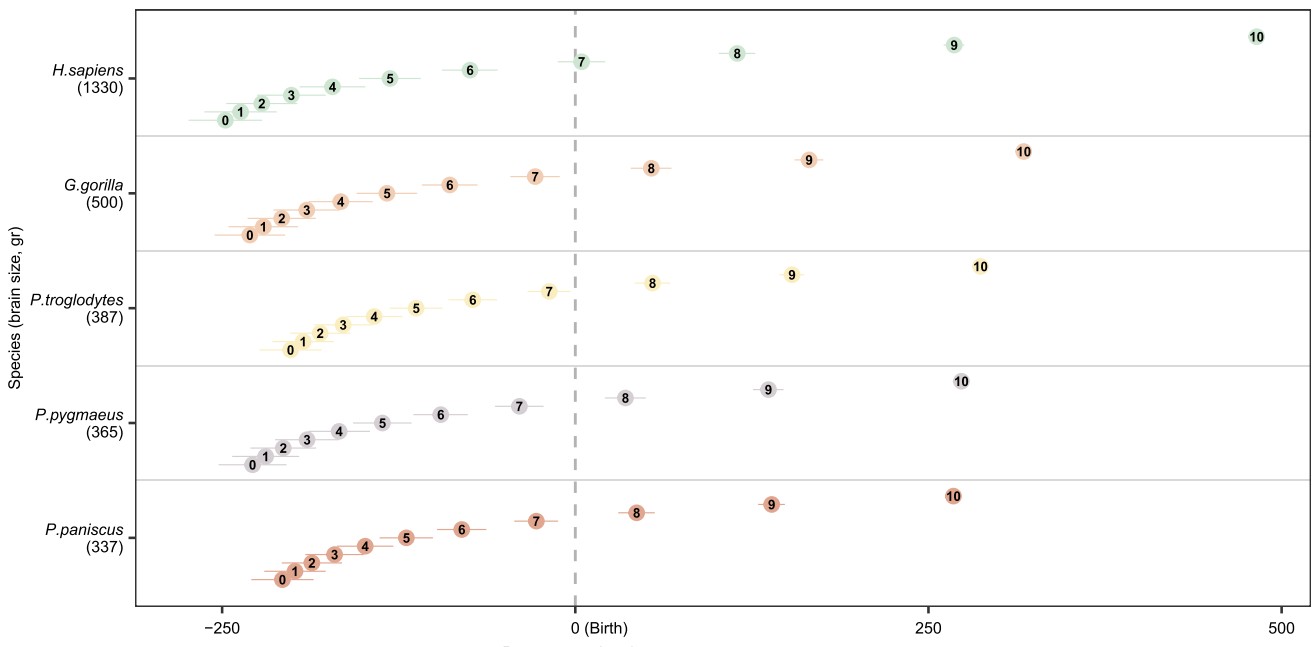

**Extended Data Fig. 4 | Comparison of the timing of neurodevelopment between humans and the great apes.** The x-axis shows the day after conception at which events with a given event score happen in each species. The y-axis shows species-specific average brain size and it is not to scale. The human average brain size used in this figure is that of present-day modern humans, which is smaller than the average brain size of fossil modern humans shown in Fig. 6. The labels 0 to 10 in the plot represent Workman's event scores 0 to 1 (1 in the plot corresponds to an event score of 0.1 and so on). The event score at birth for each species is 0.742 (*P. paniscus*), 0.756 (*P. pygmaeus*), 0.728 (*P. troglodytes*), 0.738 (*G. gorilla*), and 0.695 (*H. sapiens*). When transformed to the human scale, those event scores correspond to a mean gestation length of 321 days (10.7 months), which is the gestation length that would correspond to humans if they were born at the same neurodevelopmental stage of the other great apes.

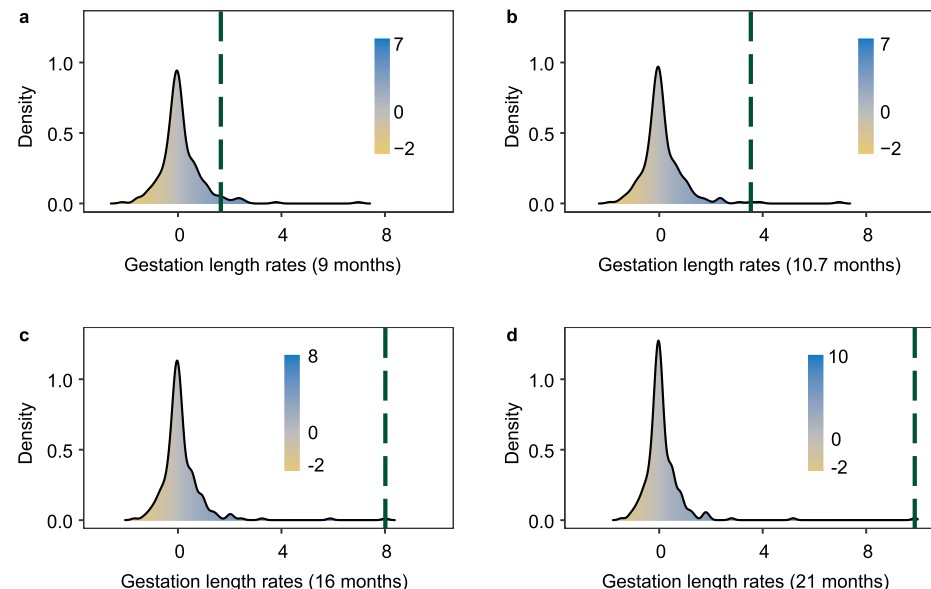

**Extended Data Fig. 5 | Evolutionary rates for human gestation length.** Comparison of the human evolutionary rate (dark green dashed line) with the distribution of evolutionary rates across mammals for a typical human gestation length of 275 days (9 months, **a**), and for hypothetical gestation lengths of 321 days (10.7 months, **b**), 480 days (16 months, **c**), and 630 days (21 months, **d**).

**Extended Data Table 1 | Body and brain proportion at birth**

| | Body proportion at birth | | Brain proportion at birth | |
|---|---|---|---|---|
| | Mean | Range | Mean | Range |
| Artiodactyls | 7.67% | 1.07%-17.42% | 43.58% | 14.36%-75.56% |
| Carnivorans | 4.03% | 0.22%-10.95% | 24.31% | 1.75%-72.73% |
| Primates | 7.31% | 2.32%-15.29% | 48.31% | 22.54%-75.34% |
| Rodents | 6.04% | 0.70%-16.38% | 25.30% | 5.36%-58.88% |
| Others | 8.81% | 1.40%-20.63% | 41.97% | 8.65%-80.72% |
| | | | | |
| Perissodactyls | 11.07% | 3.11%-15.31% | 57.98% | 51.68%-64.17% |
| Cetaceans | 9.36% | 2.91%-17.42% | 46.59% | 40.18%-53.33% |
| Pinnipeds & bears | 5.63% | 0.22%-10.95% | 36.84% | 1.75%-72.73% |
| Hominids | 4.27% | 2.32%-6.03% | 42.41% | 22.54%-57.81% |
| Elephants | 4.02% | 3.65%-4.40% | 34.67% | 32.51%-36.83% |

Mean values and ranges of variation of body and brain proportions at birth observed in the four best represented orders of mammals and in the five clades including big-brained species.

**Extended Data Table 2 | Highest evolutionary rates across the mammalian phylogeny**

| | Neonatal brain proportion | | Neonatal body proportion | | Adult absolute brain size | | Neonatal absolute brain size | | Gestation length | |
|---|---|---|---|---|---|---|---|---|---|---|
| | Decrease (more altricial) | Increase (more precocial) | Decrease (more altricial) | Increase (more precocial) | Decrease | Increase | Decrease | Increase | Decrease | Increase |
| 1st | *Homo sapiens* **(-5.94)** | Node 158 (3.99) | *Orcinus orca* (-4.76) | *Stenella attenuata* (4.46) | Node 260 (-1.47) | Node 147 (8.65) | *Stenella attenuata* (-1.84) | Node 147 (8.33) | Node 209 (-2.14) | Node 147 (6.97) |
| 2nd | *Papio papio* (-2.06) | *Macaca fuscata* (3.97) | *Stenella coeruleoalba* (-2.80) | Node 203 (3.24) | Node 208 (-1.47) | Node 151 (7.62) | Node 260 (-1.35) | *Orcinus orca* (5.66) | Node 226 (-1.64) | Node 158 (3.79) |
| 3rd | Node 256 (-2.06) | *Hylobates lar* (1.72) | *Chlorocebus pygerythrus* (-2.07) | *Chlorocebus aethiops* (2.52) | *Stenella attenuata* (-1.40) | *Orcinus orca* (5.24) | Node 208 (-1.29) | Node 151 (5.64) | Node 212 (-1.57) | Node 200 (2.59) |
| 4th | Node 162 (-1.91) | *Trachypithecus obscurus* (1.69) | *Gorilla gorilla* (-1.86) | *Cervus nippon* (2.43) | *Stenella coeruleoalba* (-1.17) | *Homo sapiens* **(4.22)** | Node 203 (-1.05) | Node 158 (4.51) | Node 164 (-1.52) | *Physeter catodon* (2.43) |
| 5th | *Macaca mulatta* (-1.82) | *Gorilla gorilla* (1.64) | Node 162 (-1.68) | *Tursiops truncatus* (1.53) | *Node 203* (-1.04) | *Physeter catodon* (3.77) | *Ursus arctos* (-0.94) | *Physeter catodon* (3.77) | Node 215 (-1.50) | *Orcinus orca* (2.39) |
| 6th | *Rusa unicolor* (-1.73) | *Orcinus orca* (1.58) | *Otolemur crassicaudatus* (-1.57) | *Homo sapiens* **(1.41)** | Node 198 (-0.90) | Node 254 (2.30) | Node 226 (-0.91) | Node 172 (2.50) | Node 155 (-1.29) | Node 172 (2.30) |
| 7th | *Stenella attenuata* (-1.57) | *Leptonychotes weddellii* (1.57) | *Rusa unicolor* (-1.47) | *Panthera pardus* (1.35) | Node 155 (-0.80) | Node 200 (2.24) | *Macaca maura* (-0.90) | Node 200 (2.44) | *Talpa europaea* (-1.28) | Node 146 (2.24) |
| 8th | *Otolemur crassicaudatus* (-1.29) | *Lama glama* (1.44) | *Cervus elaphus* (-1.21) | Node 158 (1.21) | *Hexaprotodon liberiensis* (-0.80) | Node 240 (2.10) | Node 209 (-0.90) | Node 239 (2.41) | Node 213 (-1.21) | Node 253 (2.05) |
| 9th | *Cebus capucinus* (-1.23) | *Macaca fascicularis* (1.41) | *Ateles fusciceps* (-1.02) | *Callorhinus ursinus* (1.18) | Node 169 (-0.77) | Node 255 (1.90) | *Talpa europaea* (-0.87) | Node 240 (2.08) | *Chlorocebus aethiops* (-1.18) | Node 162 (1.87) |
| 10th | *Pan troglodytes* (-1.18) | *Macaca nemestrina* (1.40) | *Panthera leo* (-1.00) | Node 267 (1.15) | *Phocoena phocoena* (-0.73) | Node 239 (1.85) | *Phocoena phocoena* (-0.83) | *Tursiops truncatus* (1.70) | Node 144 (-1.16) | Node 161 (1.75) |

Top 10 rates to increase and decrease the values of brain proportion at birth, body proportion at birth, absolute adult brain size, absolute neonatal brain size, and gestation length as shown in Fig. 2. Nodes can be identified in Extended Data Fig. 1. When nodes are listed, the relevant evolutionary rate is the rate of the branch leading to that node.

**Extended Data Table 3 | Evaluation of the values driving altriciality-precociality in human brain and body size with respect to nonhuman primates**

| | Ancestral (*Homo-Pan* LCA) | | Descendant (*Homo sapiens*) | | Descendant grade v ancestral grade | | | | | | |
|---|---|---|---|---|---|---|---|---|---|---|---|
| | Adult | Neo | Adult | Neo | Adult diff. | Neo diff. | Adult/ Neo | CI lower bound | CI upper bound | Diff. min. expectation | Diff. max. expectation |
| Brain | 6.10 | 5.28 | 7.19 | 5.70 | 1.09 | 0.42 | 2.58 | 0.91 | 1.08 | 1.67 | 1.50 |
| Body | 10.63 | 7.51 | 10.99 | 8.11 | 0.36 | 0.60 | 0.60 | 1.13 | 1.41 | -0.53 | -0.81 |

Descendant values observed in *H. sapiens* are compared with ancestral values inferred in the *Pan-Homo* last common ancestral species (LCA) to calculate neonatal and adult differences (Adult diff. and Neo diff.) and a scaling ratio between them (Adult/Neo). This ratio is then compared with the upper and lower bound of the confidence interval of the nonhuman primate scaling relationship (CI lower bound and CI upper bound). A value higher than 0 for 'Diff. max. expectation', as observed for brain size, indicates more change in adult brain size relative to neonatal brain size than the upper bound expectation of the ancestral grade. In other words, this means that the human adult brain size is larger than expected with respect to nonhuman primates and that this drives the relationship between neonatal and adult brain size. A value lower than 0 for 'Min. diff. expectation', as observed for body size, indicates less change in adult body size relative to neonatal body size than the lower bound expectation of the ancestral grade. In other words, this means that neonatal body size is larger than expected in humans with respect to nonhuman primates and that this drives the relationship between neonatal and adult body size.

**Extended Data Table 4 | Variables used in the analysis of the timing of neurodevelopment across hominin evolution**

| Species | ECV (cc) | Brain mass (gr) | PGR brain (gr/day) | Neo body mean (gr) | Neo body min (gr) | Neo body max (gr) | Gest mean (days) | Gest min (days) | Gest max (days) | Adj gest mean (days) | Adj gest min (days) | Adj gest max (days) |
|---|---|---|---|---|---|---|---|---|---|---|---|---|
| *Ar. ramidus* | 325 | 324 | 6.71 | 1129 | 1028 | 1265 | 168 | 153 | 189 | 245 | 223 | 274 |
| *Au. afarensis* | 445 | 441 | 7.63 | 1520 | 1384 | 1702 | 199 | 181 | 223 | 251 | 228 | 281 |
| *Au. afarensis* | 462 | 457 | 7.74 | 1540 | 1402 | 1724 | 199 | 181 | 223 | 251 | 228 | 281 |
| *P. boisei* | 493 | 487 | 7.93 | 1623 | 1478 | 1818 | 205 | 186 | 229 | 253 | 230 | 283 |
| *P. robustus* | 515 | 508 | 8.06 | 1697 | 1545 | 1901 | 211 | 192 | 236 | 254 | 231 | 284 |
| *H. habilis* | 610 | 600 | 8.55 | 1880 | 1712 | 2106 | 220 | 200 | 246 | 256 | 233 | 287 |
| *H. ergaster* | 801 | 783 | 9.35 | 2005 | 1826 | 2246 | 214 | 195 | 240 | 254 | 231 | 284 |
| *H. erectus* | 941 | 916 | 9.83 | 2448 | 2229 | 2742 | 249 | 227 | 279 | 262 | 238 | 293 |
| *H. heidelbergensis* | 1265 | 1222 | 10.7 | 3125 | 2846 | 3500 | 292 | 266 | 327 | 270 | 246 | 302 |
| *H. neanderthalensis* | 1487 | 1431 | 11.17 | 3514 | 3200 | 3936 | 315 | 286 | 352 | 275 | 250 | 308 |
| *H. sapiens* | 1498 | 1442 | 11.58 | 3570 | 3250 | 3998 | 308 | 281 | 345 | 274 | 249 | 307 |

Variable abbreviations: ECV: Endocranial volume; Brain mass: brain weight in grams; PGR brain: Prenatal growth rate based on endocranial volume (from ref. 27); Neo body mean: estimated neonatal body mass according to the intermediate model in ref. 18; Neo body min: estimated minimum neonatal body mass according to the human model in ref. 18; Neo body max: estimated maximum neonatal body mass according to the ape model in ref. 18; Gest mean: estimated mean gestation length based on Monson and colleagues' model and on the mean neonatal body mass; Gest min: estimated minimum gestation length based on the minimum neonatal body mass; Gest max: estimated maximum gestation length based on the maximum neonatal body mass; Adj gest mean: adjusted mean gestation length; Adj gest min: adjusted minimum gestation length; Adj gest max: adjusted maximum gestation length.

## Extended Data Table 5 | Pre- or postnatal occurrence of neurodevelopmental events in fossil hominins

| Event number in figure 6b | Event score | Event description | Event location | Prenatal occurrence (based on range) | Perinatal occurrence (based on range) | Postnatal occurrence (based on range) | Prenatal occurrence (based on mean) | Postnatal occurrence (based on mean) |
|---|---|---|---|---|---|---|---|---|
| 1 | 0.675 | Fornix myelination onset | Limbic system | HEI, ERE, ERG, HAB, ROB, BOI, AFR, AFA, RAM | SAP, NEA | None | All | None |
| 2 | 0.677 | Anterior commissure myelination onset | Limbic system | HEI, ERE, ERG, HAB, ROB, BOI, AFR, AFA, RAM | SAP, NEA | None | All | None |
| 3 | 0.68 | Lateral geniculate nucleus myelination onset | Thalamus | ERE, ERG, HAB, ROB, BOI, AFR, AFA, RAM | SAP, NEA, HEI | None | All | None |
| 4 | 0.687 | Cingulum myelination onset | Cortex | ERE, ERG, HAB, ROB, BOI, AFR, AFA, RAM | SAP, NEA, HEI | None | All | None |
| 5 | 0.689 | Mammillothalamic tract myelination onset | Limbic system | ERG, HAB, ROB, BOI, AFR, AFA, RAM | SAP, NEA, HEI, ERE | None | All | None |
| 6 | 0.692 | Internal capsule myelination onset | Cortex | HAB, ROB, BOI, AFR, AFA, RAM | SAP, NEA, HEI, ERE, ERG | None | HEI, ERE, ERG, HAB, ROB, BOI, AFR, AFA, RAM | SAP, NEA |
| 7 | 0.699 | Hippocampus myelination onset | Limbic system | HAB, ROB, BOI, AFR, AFA, RAM | SAP, NEA, HEI, ERE, ERG | None | ERE, ERG, HAB, ROB, BOI, AFR, AFA, RAM | SAP, NEA, HEI |
| 8 | 0.7 | Fasciculus retroflexus myelination onset | Limbic system | HAB, ROB, BOI, AFR, AFA, RAM | SAP, NEA, HEI, ERE, ERG | None | ERE, ERG, HAB, ROB, BOI, AFR, AFA, RAM | SAP, NEA, HEI |
| 9 | 0.703 | Stria terminalis myelination onset | Limbic system | HAB, ROB, BOI, AFR, AFA, RAM | SAP, NEA, HEI, ERE, ERG | None | ERE, ERG, HAB, ROB, BOI, AFR, AFA, RAM | SAP, NEA, HEI |
| 10 | 0.715 | Striatum myelination onset | Striatum | BOI, AFR, AFA, RAM | HEI, ERE, ERG, HAB, ROB | SAP, NEA | HAB, ROB, BOI, AFR, AFA, RAM | SAP, NEA, HEI, ERE, ERG |
| 11 | 0.722 | Corpus callosum body myelination onset | Cortex | RAM | ERE, ERG, HAB, ROB, BOI, AFR, AFA | SAP, NEA, HEI | HAB, ROB, BOI, AFR, AFA, RAM | SAP, NEA, HEI, ERE, ERG |
| 12 | 0.732 | Splenium myelination onset | Cortex | RAM | HAB, ROB, BOI, AFR, AFA | SAP, NEA, HEI, ERE, ERG | ROB, BOI, AFR, AFA, RAM | SAP, NEA, HEI, ERE, ERG, HAB |
| 13 | 0.77 | Plasticity/OD (ocular dominance) critical period–start | Cortex | None | RAM | SAP, NEA, HEI, ERE, ERG, HAB, ROB, BOI, AFR, AFA | None | All |

Only the events that differ in their pre- or postnatal occurrence across the hominin fossil record are represented, excluding those related to increases in brain size, to variation in sensory organs, and those categorized as behavioural responses of the whole organism[23]. For the range-based estimates, perinatal events are the ones that may have happened pre- or postnatally depending on the exact gestation length typical of each species. Species are listed in the pre- or postnatal columns only if a pre- or postnatal occurrence is inferred for that event for the whole range of possible gestation lengths. For the mean-based estimates, only the pre- or postnatal occurrence of each event at the mean estimate of species-specific gestation length is considered. Species abbreviations: RAM: *Ar. ramidus*; AFA: *Au. afarensis*; AFR: *Au. africanus*; BOI: *P. boisei*; ROB: *P. robustus*; HAB: *H. habilis*; ERG: *H. ergaster*; ERE: *H. erectus*; HEI: *H. heidelbergensis*; NEA: *H. neanderthalensis*; SAP: *H. sapiens*.

# Reporting Summary

## Statistics

For all statistical analyses, confirm that the following items are present in the figure legend, table legend, main text, or Methods section.

| n/a | Confirmed | |
|---|---|---|
| ☐ | ☒ | The exact sample size (*n*) for each experimental group/condition, given as a discrete number and unit of measurement |
| ☐ | ☒ | A statement on whether measurements were taken from distinct samples or whether the same sample was measured repeatedly |
| ☐ | ☒ | The statistical test(s) used AND whether they are one- or two-sided *Only common tests should be described solely by name; describe more complex techniques in the Methods section.* |
| ☐ | ☒ | A description of all covariates tested |
| ☐ | ☒ | A description of any assumptions or corrections, such as tests of normality and adjustment for multiple comparisons |
| ☐ | ☒ | A full description of the statistical parameters including central tendency (e.g. means) or other basic estimates (e.g. regression coefficient) AND variation (e.g. standard deviation) or associated estimates of uncertainty (e.g. confidence intervals) |
| ☐ | ☒ | For null hypothesis testing, the test statistic (e.g. *F*, *t*, *r*) with confidence intervals, effect sizes, degrees of freedom and *P* value noted *Give P values as exact values whenever suitable.* |
| ☐ | ☒ | For Bayesian analysis, information on the choice of priors and Markov chain Monte Carlo settings |
| ☒ | ☐ | For hierarchical and complex designs, identification of the appropriate level for tests and full reporting of outcomes |
| ☐ | ☒ | Estimates of effect sizes (e.g. Cohen's *d*, Pearson's *r*), indicating how they were calculated |

*Our web collection on statistics for biologists contains articles on many of the points above.*

## Software and code

Policy information about availability of computer code

| Data collection | Data were compiled from the literature, and no specialized software was used. |
|---|---|
| Data analysis | Data analysis was carried out in R version 4.1.3 and BayesTraits V3 (available at http://www.evolution.rdg.ac.uk/BayesTraits.html). pANCOVA analyses were carried out using functions from the R package 'evomap', which is available at https://github.com/JeroenSmaers/evomap. Other standard packages for phylogenetic analysis (such as 'phytools', 'ape' and 'geiger') and plotting ('ggplot2', 'smplot2') were used. |

For manuscripts utilizing custom algorithms or software that are central to the research but not yet described in published literature, software must be made available to editors and reviewers. We strongly encourage code deposition in a community repository (e.g. GitHub). See the Nature Portfolio guidelines for submitting code & software for further information.

## Data

Policy information about availability of data

All manuscripts must include a data availability statement. This statement should provide the following information, where applicable:

- Accession codes, unique identifiers, or web links for publicly available datasets
- A description of any restrictions on data availability
- For clinical datasets or third party data, please ensure that the statement adheres to our policy

Datasets and scripts used to carry out this study are available through the link 10.6084/m9.figshare.22242724.

## Human research participants

Policy information about studies involving human research participants and Sex and Gender in Research.

| | |
|---|---|
| Reporting on sex and gender | N/A |
| Population characteristics | N/A |
| Recruitment | N/A |
| Ethics oversight | N/A |

Note that full information on the approval of the study protocol must also be provided in the manuscript.

# Field-specific reporting

Please select the one below that is the best fit for your research. If you are not sure, read the appropriate sections before making your selection.

☐ Life sciences  ☐ Behavioural & social sciences  ☒ Ecological, evolutionary & environmental sciences

For a reference copy of the document with all sections, see nature.com/documents/nr-reporting-summary-flat.pdf

# Ecological, evolutionary & environmental sciences study design

All studies must disclose on these points even when the disclosure is negative.

| | |
|---|---|
| Study description | The study includes and in-depth analysis of data on neonatal and adult brain and body size across 140 species of placental mammals. The study includes a comparison of the proportion of neonatal to adult brain and body size across the major orders of mammals. It also includes the measurement of branch-specific evolutionary rates across the mammalian phylogeny, as well as a comparison of the scaling relationships between neonatal and adult values between orders based on phylogenetic ANCOVAs. The final part of our study includes an inference of whether key neurodevelopmental events happened pre- or postnatally across different hominin species, which is based on published models by Workman et al (2013). |
| Research sample | The studied sample includes 140 species of placental mammals, for which data were obtained on neonatal brain size, adult brain size, neonatal body size, adult body size, gestation length, and generation time. Data were compiled from the literature using the sources referenced in the 'Methods' section of the manuscript. Data were obtained for all the species of placental mammals that we were able to find in the literature. |
| Sampling strategy | Sample size was not predetermined, but all species with available data were included in our study. Only one species-specific value was included for each species, as it was not possible to obtain multiple values for each species. |
| Data collection | Data were compiled by A.G.-R. and C.N. When available, tabulated data were obtained from the supplementary information files of relevant publications. Data from older papers were obtained from the paper and copied into our general compilation. |
| Timing and spatial scale | The initial compilation of data was carried out by A.G.-R. in 2016. In 2020, C.N. checked the initial dataset based on an independent literature review, and made amendments as necessary. |
| Data exclusions | Species were included in the study only if data were available for all the variables of interest (neonatal brain size, neonatal body size, adult brain size, and adult body size). In the few cases where we could not find data on gestation length or generation time for a given species, the missing value was replaced by that corresponding to the closest sister species within the same genus. |
| Reproducibility | We checked the reproducibility of our rate analyses by calculating ancestral values using a variable rates rjBM approach and a mvBM approach (function available in the R package 'evomap'), obtaining very similar results regarding human rates. We also repeated these analyses using the residuals of a PGLS regression between neonatal and adult values and comparing the measured rates with those obtained when measuring rates for the proportion of brain and body size at birth. We also recalculated evolutionary rates using a comparison of the observed amounts of change per branch with those obtained when simulating evolution over the mammalian phylogeny. Those rates show a very high correlation with the rates obtained by comparison of the observed amount of change per branch with the expected amount of change per branch based on branch lengths. |
| Randomization | The classification of mammalian species within the four major orders we have used in our study is well established and randomization is not relevant. |
| Blinding | The values used in our study are associated with the species they belong to, and they were reviewed based on this species identity. Therefore, blinding was not possible. |

Did the study involve field work?   ☐ Yes   ☒ No

# Reporting for specific materials, systems and methods

We require information from authors about some types of materials, experimental systems and methods used in many studies. Here, indicate whether each material, system or method listed is relevant to your study. If you are not sure if a list item applies to your research, read the appropriate section before selecting a response.

## Materials & experimental systems

| n/a | Involved in the study |
|-----|------------------------|
| ☒ | ☐ Antibodies |
| ☒ | ☐ Eukaryotic cell lines |
| ☒ | ☐ Palaeontology and archaeology |
| ☒ | ☐ Animals and other organisms |
| ☒ | ☐ Clinical data |
| ☒ | ☐ Dual use research of concern |

## Methods

| n/a | Involved in the study |
|-----|------------------------|
| ☒ | ☐ ChIP-seq |
| ☒ | ☐ Flow cytometry |
| ☒ | ☐ MRI-based neuroimaging |

