## [Peer Review File · Nature Ecology & Evolution]

Peer Review Information

Journal: Nature Ecology & Evolution

Manuscript Title: The evolution of human altriciality and brain development in comparative context

Corresponding author name(s): Aida Gómez-Robles

Editorial Notes:

Reviewer Comments & Decisions:

Decision Letter, initial version:

6th April 2023

Dear Dr Gomez-Robles,

I am writing to you in the temporary absence of my colleague Dr Nic Eoin.

Your Article, "The evolution of human altriciality in comparative context" has now been seen by three reviewers. You will see from their comments copied below that while they find your work of considerable potential interest, they have raised quite substantial concerns that must be addressed. In light of these comments, we cannot accept the manuscript for publication, but would be very interested in considering a revised version that addresses these serious concerns.

We hope you will find the reviewers' comments useful as you decide how to proceed. If you wish to submit a substantially revised manuscript, please bear in mind that we will be reluctant to approach the reviewers again in the absence of major revisions.

* Include a "Response to reviewers" document detailing, point-by-point, how you addressed each referee comment. If no action was taken to address a point, you must provide a compelling argument. This response will be sent back to the referees along with the revised manuscript.

* If you have not done so already we suggest that you begin to revise your manuscript so that it conforms to our Article format instructions at <http://www.nature.com/natecolevol/info/final-submission>. Refer also to any guidelines provided in this letter.

2Please use the link below to submit a revised paper:

[REDACTED]

If you wish to submit a suitably revised manuscript we would hope to receive it within 6 months. If you cannot send it within this time, please let us know. We will be happy to consider your revision so long as nothing similar has been accepted for publication at Nature Ecology & Evolution or published elsewhere.

Nature Ecology & Evolution is committed to improving transparency in authorship. As part of our efforts in this direction, we are now requesting that all authors identified as 'corresponding author' on published papers create and link their Open Researcher and Contributor Identifier (ORCID) with their account on the Manuscript Tracking System (MTS), prior to acceptance. This applies to primary research papers only. ORCID helps the scientific community achieve unambiguous attribution of all scholarly contributions. You can create and link your ORCID from the home page of the MTS by clicking on 'Modify my Springer Nature account'. For more information please visit www.springernature.com/orcid.

Thank you for the opportunity to review your work.

[REDACTED]

Reviewers' comments:

Reviewer #1 (Remarks to the Author):

I reviewed this manuscript thoroughly. I much enjoyed reading this manuscript and found it very interesting. It is well-written, the analyses are rigorous, and the focus and content of the manuscript should be of interest to a broad readership as it deals not only with human evolution, but also with brain evolution, a set of key human traits (helplessness at birth and relatively low % of adult brain size achieved at birth) and situates all these in a comparative mammalian context. I believe it is suitable for the scope of NEE and I can recommend this manuscript for publication after the authors have a chance to address the below suggestions, comments and questions and make at least some changes accordingly. Please read my comments and suggestions below in the context and tone of this positive evaluation.

GENERAL COMMENTS / ISSUES

2- In the Introduction, the term altriciality is immediately introduced but not clearly defined. If anything, L.32-34 provides a general, though not very specific, definition of altriciality in relation to the human condition, and it is not unambiguously only about neural development. This is in fact appropriate as most biologists will think of a number of other traits when it comes to altriciality vs. precociality (eyes/ears closed, ability to move around independently and support the body with the limbs, etc.). But soon after that and throughout the rest of the manuscript, altriciality is rather implicitly equated with the proportion of adult brain size achieved at birth, or at least with brain altriciality. In L.88-90, the authors are a bit more explicit and write that a common measure of degree of altriciality/precociality is this neonatal brain proportion, though no references are given; this feels a bit reductionist with regard to the neuro-sensory-behavioral complexity of the altriciality-precociality spectrum. Then in L. 132-133, the authors specifically talk about "brain altriciality", which could read as one particular but non-exhaustive aspect of altriciality. In L. 315-318 in the Discussion, the authors once again seem to acknowledge that there are different ways in which a mammal can be altricial or precocial, opening the possibility that depending on which measure one uses, the patterns among placentals - or that of humans in relation to other mammals - would change, but this is not really addressed. I think it would be helpful if the authors are more explicit about the particular definition and scope of altriciality that they employ in their work and that the part that they focus on does not capture the entire neuro-sensory-behavioral complexity of altriciality and precociality. Or, in case the authors wish to make the case that altriciality/precociality can be reduced to brain proportion at birth or number of neural developmental events achieved at birth (do the authors think these are the same thing by the way?), this needs to be substantiated.

- Similarly, the authors link the concept of plasticity to brain altriciality and the timing of neurodevelopment, but brain plasticity is not defined/explained and the connections between these concepts are not made explicit or very clear throughout the manuscript. Yet, inferences about plasticity seem a main goal of the study as implied in L. 79-82 and L. 259-261, and elsewhere. The manuscript would really benefit from clarification in this regard.

- The second aim and second set of analyses (essentially the work based on Workman et al.'s work) needs to be explained better/in more detail starting in the Introduction and including the Results. I could not understand and interpret the second set of analyses in the Results without having read the Methods. Specifically, I found the paragraph in L. 68-82 unclear with regard to what exactly the authors are aiming to achieve and what they are referring to when they say "ontogenetic series", "developmental pattern", "shift", "timing of neurodevelopment" etc. See also my comment above about a definition of altriciality, because in L. 75 the authors seem to have an implicit definition of altriciality in mind when they say "altriciality might have evolved in earlier hominins", and at this point it is not clear how they will measure or indeed define it. Reading this paragraph (L. 68-82) again after finishing the whole manuscript (and to understand the results, including Extended Data Fig. 3, one needs to read the methods), I infer that the authors use altriciality to mean - in this paragraph at least - a specific and comparatively early stage of neural development at birth, when some but not all neurodevelopmental events have already happened, and referring specifically to Workman et al.'s work. The manuscript would benefit from this second aim to be (a) explicitly defined as a second aim in the Intro, and (b) to be explained in more detail (Intro + Results) with regard to the data that are used and what aspect of altriciality they measure, much in the same way that they do for brain and

3body size proportions for the first aim.

SPECIFIC / MINOR COMMENTS AND QUESTIONS

** Abstract

L. 21: "Based on our data" rather than "Based on our results".

L. 26-28: I didn't quite know how to interpret this last sentence. Seeing as how the authors stated their main findings earlier in the abstract, I expected this last sentence to be some final, overarching conclusion but it feels a bit vague instead, as if the authors are not yet divulging how their study shed light on this hypothesis. See also my earlier, general comment on plasticity.

** Introduction

- L. 35: see also Rosenberg 2021 (<https://doi.org/10.1146/annurev-anthro-111819-105454>) for the association with the development of social and emotional relationships.

- L. 46-47: better to say that selection favored neonates with smaller heads/brains *at the time of birth*, otherwise it is a bit ambiguous. Rosenberg 1992 is a good reference here but the classical reference is Washburn 1960 (10.1038/scientificamerican0960-62), which the authors may wish to add.

- L. 51-52: "it can be inferred that [human altriciality] is associated with substantial adaptive advantages, as otherwise selection would likely have acted against it." I encourage a rephrasing, because right now it reads a bit adaptationist, as if altriciality in humans must have had adaptive advantages in its own right, irrespective of the constraints imposed by the pelvis and/or maternal metabolism, but this is hard to prove. It may have had adaptive advantages in its own right (increased plasticity, opportunities for social learning, etc.) but even the constraints imposed by maternal anatomy and/or physiology favors altriciality as a compromise solution between fetal development and obstetric/metabolic sufficiency. Based on what is written, I presume the latter is what the authors mean, but it is ambiguous.

L. 55: add "also" to "...could [also] explain why a seemingly..." because these explanations are not mutually exclusive.

L. 60: "To assess how altriciality evolved in humans", I find the "how" here rather vague (and the rest of the sentence does not make it clearer).

L. 63-64: the authors could cite their existing ref Jones et al. 2009 (panTHERIA database) for body size variation in mammals.

L. 72: "species-specific ontogenetic series" of what?

4** Results

- In L. 93-94, the authors state that "body proportion at birth appears to be highly constrained across mammals." Although body size proportions are not the primary focus of the manuscript, this statement is nonetheless a bit misleading/inaccurate based on the authors' own data/results and the published body of work on body proportion at birth. Their study relies on matching data for neonatal brain and body size proportions across all taxa studied, which limits the potential sample size (and at N=140 the authors have done well) and thus misses out on a lot of published neonatal body size data. The authors claim that, at <25%, body proportions are highly conserved in mammals, but a range of ~25% is arguably not little. They furthermore state there are minor differences in body proportion across clades (L.91-93), but this ignores the variation within clades (visible in Fig. 1b but not explicated by the authors). Also, bats give birth to neonates/fetuses up to a whopping 45% of maternal body size and these data have previously been presented in the context of human evolution, altriciality and pelvic evolution (Grunstra et al. 2019 <https://doi.org/10.1002/ajhb.23227>). In fact, for 2 of the bat species used by the authors, *Nyctalus noctula* and *Plecotus auritus*, the adult body proportions achieved at birth are 19.5% and 25%, resp., based on the PanTHERIA data. However, Extended Data Table 1 shows a maximum value of 9.99% for the category that includes the bats, and this must therefore be wrong or based on inaccurate data (for reference, Kurta & Kunz 1987 Symp. zool. Soc. Lond., no. 57, even report 19.7% and 30.1% for these 2 species). *Please revise.* Most bat species do not have associated neonatal and adult brain size data, precluding their inclusion in this study, but the statement that body size proportions are "highly constrained among mammals" (well, "placentals" because marsupials are another story) I feel is nonetheless incorrect given all of the above. The authors want to highlight the "contrast" (L. 98) with the variation in brain proportion at birth (where variation within orders *is* interpreted), but they don't need to throw the variation in body size proportions under the proverbial bus in order to make this point.

- L. 150-151 & Fig. 3b: the authors state that unlike in all other well-sampled orders, there is no relationship between the evolutionary rates in body and brain proportions in primates, but it looks like there may in fact be a (weak, negative) relationship in this group if it wasn't for humans. Did the authors calculate the correlation in primates without humans? Because outliers, which humans clearly are in this plot, can have strong effects on correlations and regressions, especially if they are extreme values (as humans are here). I would not characterize an entire clade based on the undue influence of one data point.

- Similarly, but in the opposite way, in the next paragraph (L. 153-159), primates are described to have a particularly strong negative correlation between absolute adult brain size and brain altriciality (brain proportion at birth), which also seems due to the influence of the human data point in the top left corner of the relevant panel of Fig. 3b. Of course humans' contribution to the primate pattern is legitimate as they/we are primates, but the authors should be careful when interpreting humans in their primate context, and indeed when interpreting primates in a broader placental mammalian context, when this "context" is particularly strongly influenced by humans themselves. Even just acknowledging the influence of the human data point might be sufficient here.

- L. 172-174: do these correlations exist within any of the clades?

- section starting with L. 178 & Figs. 4 & 5: I figured out that the neonatal and adult body and brain sizes are analyzed on a natural log (ln, not common log) scale, but this is not mentioned in the text nor in the legends of Figs. 4 and 5.

- L. 181: the authors report the same slope and intercept for the four best-sampled orders here, but in the legend of Fig. 4b they report a significant difference between carnivorans and primates.

- section L. 178-204: clade-specific slopes and intercepts are compared and outliers within each clade are inspected separately. This is a nice part of the analysis and the findings are interesting and illustrative. However, the authors make no mention of the fact that carnivorans include two rather distinct groups of mammals in terms, namely the terrestrial carnivorans and the aquatic pinnipeds (seals). These two groups differ markedly in where they fall on the altriciality-precociality spectrum, including litter size, all of which seems highly relevant to the focus of this paper. Indeed, I think I can see which carnivoran data points are the terrestrial ones and which the pinnipeds just by looking at the plots in Figs. 4 and 5. With regard to the "outlier analysis" in Fig. 5 I think an argument could be made to regard not the bears as outliers, but the pinnipeds as outliers to an otherwise terrestrial carnivoran pattern. It was not clear from the Methods whether the regressions/pANCOVAs are conducted on the entire sample or not, i.e. including or excluding the apparent outliers, but in case it is the latter, then pinnipeds would be equally if not more appropriate "outliers" as they represent a true grade shift within Carnivora. Also, the big cats (*Panthera* sp.) appear to be missing in Fig. 5c and 5d, and at least *Panthera leo* and *P. tigris* would cluster close to the bears, which ostensibly would have an effect on the regression line, confidence/prediction intervals and possibly on who is an outlier. This requires critical re-analysis.

- Similarly, the toothed whales are nested within artiodactyls ("Cetartiodactyla") and they are often compared to anthropoid primates for their increased intelligence, learning capacity, cultural behaviors and sociality. Interestingly, the data points (Figs. 4 and 5) suggest that whales as a whole may not deviate from the terrestrial artiodactyl pattern, possibly because all cetartiodactyls tend to give birth to a single, well-developed fetus. I think it could be interesting to point out. By contrast, suiformes are the odd ones out, which is very likely due to the fact that they alone give birth to litters >1. Again, a super brief sentence explaining this seems worthwhile and makes the most of the authors' comparative approach.

- where are the rodents in Fig. 5? Was there no apparent outlier in this group? This should at least be mentioned somewhere.

The points below relate to my major/fundamental comment above on the neurodevelopmental timing analysis. For example:

- L. 208-211: this needs more explanation.

- L. 222-225: what are these events? Providing a couple of examples would help.

- L. 231: what "range" are the authors referring to? Event scores > 0.788 or the range 0.670-0.788? And I didn't understand how "the range of event scores...probably narrower" followed from the preceding part of the sentence about variation in gestation length among hominins. (By the time I reached this part of your interesting manuscript, I felt I had to work too hard to follow the authors.)
- L. 233: "within this range", is this the same range as above?
- I know NEE employs a format where the Methods only appear at the end, but this section of the results really requires a bit more explanation of these neurodevelopmental events and how the present authors calculated whether they would occur pre- or postnatally.
- a brief explanation of the significance of myelination or its developmental timing here (Results) or in the Discussion would be appreciated.

** Discussion

- L. 259-261: "[...] our study has focused on brain development in an attempt to understand whether human altriciality has evolved as a selectively advantageous trait in association with increased brain plasticity." The authors have not sufficiently explained how brain altriciality leads to increased plasticity, nor have they argued (however briefly) that (and why) neural plasticity is selectively advantageous earlier on in the manuscript, so this sentence comes somewhat as a surprise to the reader. See my one of my earlier general comments.
- L. 287-289: I don't understand the conclusion the authors draw here; I don't follow how the sentence relates to the preceding text or what exactly the point is that the authors try to make.
- L. 335-339: Also here, as a reader I feel like I'm missing the relevant background and context explained earlier in the manuscript to (a) expect this kind of discussion and (b) to know what to do with this information.
- L. 341-350: same as above + it gets a bit technical / specific terms that the non-neural expert may not follow (e.g. "activity-dependent myelination", "brainwave rhythms").
- L. 352-360: The relevance of this paragraph is not very clear to me. (Given the above, if the authors need to shorten the manuscript due to limits on article length, this could probably be achieved in this part of the Discussion.)

** Methods

- L. 438-439: very good.
- L. 442: I presume the authors use Pearson correlations?

- L. 458: Is the expected scaling relationship established on the sample excluding the outlier? I think this should be made explicit. (See also my earlier comment in the Results.)

- L. 501-503: I would rather say "speculations" or something more careful than "general observations" when it comes to the inferences made about Neanderthal gestation length by Trinkaus, because this has since been debunked/refuted and is no longer held to be true by the community.

- L. 504-515: the authors describe how they dealt with the uncertainty in gestation length for the different fossil hominins, but maybe they can also quickly report on how the main results/outcomes of this analysis differed (or whether they differed) depending on what gestation length was used for these different species. Right now this is not transparent.

** Figures and Tables

Fig. 1: Great figure and I appreciate all the original data being displayed alongside the boxplots. Perhaps the authors can consider briefly defining "brain and body proportion" in the legend (e.g. "% adult size attained at birth"), because at this point in the manuscript the reader is not yet familiar with these terms.

Fig. 2: What do the little open and filled (black) circles on the nodes and tips represent, and what does the difference in the circle size represent? Are they scaled to the values of brain and body sizes, proportions, etc.?

Fig. 4: units of measurement / log scale missing (see my earlier comment about this).

Fig. 5: Panthera sp. (x3) are missing from plots c and d (see also my earlier comments about this fig and the outliers)

Extended Data Fig. 2: not really sure what the authors are trying to show here, so please clarify.

Extended Data Fig. 3: I found this figure challenging to interpret and understand when reading the manuscript chronologically (see also my earlier comments on general clarification of the aim and analysis of this part of the study). Among other things, I found the combination of gestation length along the Y axis and the slanting of the lines confusing; why display gestation length when you're not depicting any variation among taxa in gestation length? Why are the lines slanted? Some lines cross the vertical birth line, but it is not clear from the fig whether it's significant/meaningful that the top or bottom of the taxonomic lines crosses the vertical birth line. What determines the position of the taxonomic lines along the x axis? Without more clarification in the main text (i.e. before the methods section) and in the figure legend itself, I thought this figure was not very accessible.

** Data files and scripts

Looks like all data files and scripts for a reproduction of the analysis are there.

Reviewer #2 (Remarks to the Author):

Thanks very much to the authors for this manuscript and especially for the study that went into it. With major revisions (including additional analyses) it will be a strong contribution to human evolutionary science that will move us forward on the question of altriciality. I have organized my feedback into four categories, below: (1) The problem with the adult sample being both sexes (with no additional analysis using only females), (2) Clarity and consistency of terms/concepts, (3) Additional references, (4) Evolutionary thinking.

Thanks again, very much.

Sincerely,

Holly Dunsworth

1. The problem of the combined sex adult sample

Line 385 reveals that the neonatal measures are being compared to a combined sample of adult males and females, when they should be used in comparison to female adults exclusively--if not as the sole analysis, than as an additional one included in this paper. This is a problem that requires major revision. I'm sitting here worried that I misinterpreted line 385, given the figures have silhouettes of curvy human females (not males), which leads readers to assume all the adults in the analyses are females, only. If that's true, that's fantastic and there is no problem. But the word female is absent entirely from the paper and where "maternal" shows up, it's to describe previous work, not this analysis. And then there's line 385. So, it seems those figures with the curvy human female silhouettes are misrepresenting the actual analysis. The only acceptable analysis to my mind excludes adult males. If argued well, I could imagine how an analysis using two kinds of adult samples (one pooling the sexes and one female-only) could fly, but the analysis in this manuscript that uses only a pooled adult sample does not fly. Maternal/female adult size is the ideal comparison because of what the authors say in line 385 and because of known metabolic constraints of gestation and lactation that are reflected in/correlated with maternal size (that the authors acknowledge elsewhere in the paper). New analyses need to be included with a female-only adult sample.

2. Clarity and consistency of terms/concepts: Altriciality, Brain altriciality, brain size altriciality, brain proportion, body precociality, body proportion, etc.

There's altriciality and then there's what is under study in this paper, which is part of that phenomenon/concept but not equivalent it. Please make all this very clear from the very beginning of the paper, and then use the specific terms about what you're specifically discussing within the overall concept of altriciality throughout.

Brain size altriciality needs to be defined in the Introduction, not in the first sentence of the results. And the variable is actually called "brain proportion" elsewhere, so there's really a need for consistency and clarity on this, throughout the manuscript. When used, please call it "brain size altriciality" not just "brain altriciality" because the latter conjures other developmental phenomena beyond mere size, which are acknowledged in the paper and considered separately. Again, this choice

9of variable has got to be set up in the Intro with a stronger context for what “altriciality” is as well. The manuscript could really use a firmer foundation there.

Line 136. I believe this is a typo and is meant to say “body precociality” which should be clearly defined now because I believe it’s the first mention in the text. I’m also suggesting that you call this variable instead “body size precociality” because just “body” alone conjures motoneuronal development which is considered separately. But wait! It’s “body proportion” elsewhere. This is the same problem as outlined above.

3. Three References that need to be cited in collaboration

1. Karen R. Rosenberg. 2021. The Evolution of Human Infancy: Why It Helps to Be Helpless. *Annual Review of Anthropology* 50:1, 423-440 <https://www.annualreviews.org/doi/abs/10.1146/annurev-anthro-111819-105454>

Rosenberg’s paper offers many things including (a) a stronger basis for the Introduction of this manuscript and (b) ideas to cite about the long postnatal growth period being adaptive and potentially the target of selection itself (rather than the byproduct of constraints on gestation and pregnancy).

2. Holly Dunsworth. 2021. (Chapter 27) There is No Evolutionary “Obstetrical Dilemma”. In C. Tomori and S. Han, (Eds.), *The Routledge Handbook of Anthropology and Reproduction*. Taylor and Francis. https://digitalcommons.uri.edu/soc_facpubs/41/

This paper supports the arguments and conclusions made in this manuscript that increasing absolute adult brain size is causing decreasing relative neonatal brain size. From Dunsworth 2021: “It has long been taken for granted that as monkeys, apes, and hominins encephalized, their postnatal brain growth increased and so their relative brain size at birth decreased. No one has ever asserted that increased encephalization should occur during gestational days alone. And yet, this is what Gould’s argument implies—that in order to grow larger brains, our ancestors should have reversed primate tradition and accomplished increased brain growth in utero. To be clear, this is an assumption that human encephalization should have been unlike any other primate’s. However, if one approaches human encephalization from a comparative primate perspective, one expects that with increasing adult hominin brain size there must have been decreasing relative brain size at birth, regardless of childbirth difficulty.” And the paper also goes into the context for why human evolutionary scientists have long assumed or entertained the idea that humans should have a much longer gestation/larger percent of adult brain at birth than we do.

3. Tesla Monson et al. 2022. Teeth, prenatal growth rates, and the evolution of human-like pregnancy in later Homo. *PNAS* <https://www.pnas.org/doi/10.1073/pnas.2200689119>

This paper estimates gestation length in fossil hominins and is thus highly relevant to that portion of the manuscript.

In addition, I found the fossil hominin portion of the text, but especially the table that goes along with it, a bit hard to follow. Maybe it’s because I knew the Monson paper and so I was thrown by what I brought, myself via that paper. But re: the table: The headers on the columns should better explain what those columns contain. The figure doesn’t and shouldn’t have to do the job for them. And if there is room to convey “all the rest of the hominins” instead of “the rest” that would also help. To a

10newcomer reader seeing "Prenatal: the rest" is disorienting. I think it's as simple as editing the headers to read: Occurs prenatally, Occurs perinatally, occurs postnatally, or something like that.

4. Evolutionary thinking

Line 52. This little bit on evolutionary thinking could use a bit more nuance. As-is it's just not correct. That something observable exists (i.e. has not been selected against and eliminated) does not mean that it therefore confers a substantial adaptive advantage. What works, works. In this case specifically, older children and adults are brilliant at many things and one of them is caring for young with an extended postnatal growth period. It need not experience positive selection directly in order for it to exist.

But "moving a greater proportion of brain maturation to the postnatal period" (line 56) is not what is happening, biologically. I think it's important (though difficult to articulate) to point this out because it looks like this is being considered as a target of selection, or is being described as others' hypothesized target of selection. It cannot be. Maybe I sound like I'm splitting hairs or arguing semantics but I'm not. If "altriciality" in humans or, instead, what I would call a long postnatal developmental period (and can do so even more confidently thanks to this paper) was adaptive and was being increased in duration by selection, then that need not be accomplished by taking away gestation time, by being born earlier. It could be accomplished by lengthening the postnatal growth period. And if it occurred because of a shortening of gestation, then it's by exactly that, selection for a shorter gestation or an earlier birth, not by "moving a greater proportion of brain maturation to the postnatal period." That "moving..." bit describes how the numbers look to us, but it's not what biology would be literally doing over time. To try to convey what I mean, here's that sentence, reworded to describe the biology, instead: "Indeed, a long postnatal growth period offers the opportunity for ..."

At the start of the Discussion, in the first sentence with "humans evolved their high level of brain altriciality at a very fast rate" I had a familiar reaction to one I tried to convey above about "moving...". (note: adding size so that the variable is called "brain size altriciality" would go a long way to clarifying this paper.) Because of the results section, I can't help but assume that you're referring to your variable "brain proportion" here. Talking about humans evolving a proportion of their adult brain size at birth is a problem. That's not an actual trait of an animal. Selection doesn't "act" on the size of a neonatal brain in relation to what it's eventual adult size will be. That's a proxy we made up. That's how the field has talked about it and it's just strange. When I read that sentence I shouldn't have to substitute "long postnatal growth period" or "high rate of gestational growth" or "short gestational growth period" in my head for "high level of brain altriciality" but I am because neonatal/adult brain size at birth is not an actual trait that selection can "see". I hope I'm making sense. Of course relative brain size at birth evolves, but it evolves the way that sex differences in body size evolve: the thing itself is changing over time but not because of selection directly on the thing itself. Absolute brain size at birth, yes. Gestation length and growth rate, yes. Postnatal length and growth rate, sure. The brain proportion variable is great for study, but thinking about it like it's an adaptive trait itself is problematic. It's fine being a proxy for long postnatal growth period in the evolutionary analysis. This isn't a criticism of the study itself; it's a critique of the assumptions and thinking that this study fails to support. Maybe my suggestion, then, is that the authors acknowledge this issue and that they're using a proxy variable not an actual trait that can experience selection in

11their paper. It will help strengthen the case!

And it will really help with that hypothetical exercise against Gould's influence on human evolutionary science (see all the context in that Dunsworth 2021 paper above) starting on line 277. That (useful) exercise you undertake shows how ridiculous it has been for our field to assume that percent brain size at birth is an actual trait! But because that's not acknowledged, it's a strange read. I think readers need to know what the point of this exercise is because many are stuck in thinking that percent brain size at birth is something that could be selected and so they elevate it to the level of the traits that can and that muddies this whole endeavor. Most importantly, the summary/conclusion paragraph will be a lot stronger when edited to reflect this issue that your study is revealing to our discipline, which is much needed. In general, not just with this important aspect, this paper really deserves much clearer statement of its fresh approach and fresh insight to the question of human altriciality.

Line 98...That's an interesting range of relative neonatal brain size across orders and so it seems like a missed opportunity to consider whether those born with a higher percent of adult brain are more precocious than those born with less. I hope you can add something about that. However, if tracking down those observational data is a challenge, then that may be beyond the scope of this paper and may, instead, be acknowledged in some useful, contributing way.

Paragraph starting at line 98. I don't think the relevance of these results is considered enough in the Discussion. It seems to me that the text should say more explicitly that percent neonatal brain size isn't helpful for understanding human evolution. Right? If so, that's a revelation for the field and needs to be made clear. This links back to my comments about it not being an actual trait, but instead a proxy and in this case... it's the long postnatal period

Reviewer #3 (Remarks to the Author):

This is an interesting paper and the topic presented is likely to be of interest to a broad audience. However, I have some major concerns that preclude my recommendation for publication which I shall outline below.

The first and primary concern I have is with regards to the methods used to evaluate evolutionary rate heterogeneity. There are several methods available in the literature to do this – and the one the authors choose to use (mvBM) is one that has been criticised (Griffin & Yapuncich 2016, 2017) without clear rebuttal. Further, alternative approaches to estimating trait rate variation systematically outperform mvBM (Cooney and Thomas 2021). The citation history for the method (the authors reference 60) speaks volumes – whilst the method has been used in a few instances, it is largely cited by the authors themselves or in the context of discussing methodological approaches. I expect the authors will fight on this point, but the published criticisms of this approach are difficult to ignore.

The second major concern I have with the manuscript as it stands is the use of proportions as a standalone evolutionary entity. The issue here is that the authors are treating ratios/proportions as

12evolving quantities, using methods that assume data are distributed according to the predictions of evolutionary models. This is likely to not be the case. Selection does not “see” a proportion, but rather neonatal brain size (or body size, etc). Why not study the trait of interest whilst accounting for the other? The risk here is that by treating these quantities as the target of evolution, it introduces biases in the data – especially when studying traits that are hugely collinear such as brain and body size. By taking advantage of the approaches that allow multivariate models (which are utilized in this manuscript for pattern comparisons) not only does it reduce biases, but it also streamlines the paper.

My final concern lies with the analyses regarding fossil hominins. I’m reticent to even refer to these even really an analysis, rather a set of calculations. It is not really clear what exactly the authors seek to gain from the calculations they make nor what they are trying to say with these sections. It looks to me that they are simply trying to estimate a quantity from fossil hominins (from essentially no data) that they then simply use in a brief descriptive paragraph in the discussion. I would argue that without significantly more statistically rigorous techniques and much more in-depth analysis/discussion, these sections are not adding much – if anything – to the manuscript.

Overall, I am sorry to say that I cannot faithfully recommend this manuscript for publication in its present form.

- Randi H. Griffin, Gabriel S. Yapuncich, A critical comment on the ‘multiple variance Brownian motion’ model of Smaers et al. (2016), *Biological Journal of the Linnean Society*, Volume 121, Issue 1, 1 May 2017, Pages 223–228
- Griffin, Randi H., and Gabriel S. Yapuncich. "The independent evolution method is not a viable phylogenetic comparative method." *Plos one* 10.12 (2015): e0144147.
- Cooney, Christopher R., and Gavin H. Thomas. "Heterogeneous relationships between rates of speciation and body size evolution across vertebrate clades." *Nature ecology & evolution* 5.1 (2021): 101-110.

Author Rebuttal to Initial comments

Dear Editor,

Thank you for your feedback on our manuscript. We have extensively revised our manuscript in response to the points raised by the three reviewers. We believe that the manuscript has improved substantially based on these changes, so we would like to thank the three reviewers for their time and feedback. We provide below a point-by-point response to each of the comments raised by the three reviewers, but we would like to highlight below the most substantive changes:

1. In response to reviewer #2, body size analyses now use maternal body sizes.
2. In response to reviewer #3’s concerns, rate analyses are based now on a reversible jump Brownian motion (rjBM) approach to calculate ancestral values.

133. In response to comments from the three reviewers, the fossil analysis has been completely revamped. These analyses now use estimates of species-specific gestation lengths, and results are presented in a more intuitive way.
4. Additional analyses and clarifications have been provided in response to the extensive feedback provided by reviewer #1.

Reviewer #1 (Remarks to the Author):

I reviewed this manuscript thoroughly. I much enjoyed reading this manuscript and found it very interesting. It is well-written, the analyses are rigorous, and the focus and content of the manuscript should be of interest to a broad readership as it deals not only with human evolution, but also with brain evolution, a set of key human traits (helplessness at birth and relatively low % of adult brain size achieved at birth) and situates all these in a comparative mammalian context. I believe it is suitable for the scope of NEE and I can recommend this manuscript for publication after the authors have a chance to address the below suggestions, comments and questions and make at least some changes accordingly. Please read my comments and suggestions below in the context and tone of this positive evaluation.

We would like to thank reviewer #1 for their exhaustive and constructive feedback

GENERAL COMMENTS / ISSUES

- In the Introduction, the term altriciality is immediately introduced but not clearly defined. If anything, L.32-34 provides a general, though not very specific, definition of altriciality in relation to the human condition, and it is not unambiguously only about neural development. This is in fact appropriate as most biologists will think of a number of other traits when it comes to altriciality vs. precociality (eyes/ears closed, ability to move around independently and support the body with the limbs, etc.). But soon after that and throughout the rest of the manuscript, altriciality is rather implicitly equated with the proportion of adult brain size achieved at birth, or at least with brain altriciality. In L.88-90, the authors are a bit more explicit and write that a common measure of degree of altriciality/precociality is this neonatal brain proportion, though no references are given; this feels a bit reductionist with regard to the neuro-sensory-behavioral complexity of the altriciality-precociality spectrum. Then in L. 132-133, the authors specifically talk about "brain altriciality", which could read as one particular but non-exhaustive aspect of altriciality. In L. 315-318 in the Discussion, the authors once again seem to acknowledge that there are different ways in which a mammal can be altricial or precocial, opening the possibility that depending on which measure one uses, the patterns among placentals - or that of humans in relation to other mammals - would change, but this is not really addressed. I think it would be

14helpful if the authors are more explicit about the particular definition and scope of altriciality that they employ in their work and that the part that they focus on does not capture the entire neuro-sensory-behavioral complexity of altriciality and precociality. Or, in case the authors wish to make the case that altriciality/precociality can be reduced to brain proportion at birth or number of neural developmental events achieved at birth (do the authors think these are the same thing by the way?), this needs to be substantiated.

The new version of the manuscript starts with a clear definition of altriciality and precociality and the traits that are associated with each of these developmental patterns. After that, we explain that the first part of the manuscript focuses on one particular aspect of altriciality (% of adult brain size at birth) because of the historical importance of this variable in discussions about human altriciality. Later in the manuscript, we discuss how this metric correlates with measurements of altriciality based on neurodevelopmental stage at birth.

- Similarly, the authors link the concept of plasticity to brain altriciality and the timing of neurodevelopment, but brain plasticity is not defined/explained and the connections between these concepts are not made explicit or very clear throughout the manuscript. Yet, inferences about plasticity seem a main goal of the study as implied in L. 79-82 and L. 259-261, and elsewhere. The manuscript would really benefit from clarification in this regard.

Brain plasticity and its possible association with altriciality are discussed explicitly in the new version of the manuscript.

- The second aim and second set of analyses (essentially the work based on Workman et al.'s work) needs to be explained better/in more detail starting in the Introduction and including the Results. I could not understand and interpret the second set of analyses in the Results without having read the Methods. Specifically, I found the paragraph in L. 68-82 unclear with regard to what exactly the authors are aiming to achieve and what they are referring to when they say "ontogenetic series", "developmental pattern", "shift", "timing of neurodevelopment" etc. See also my comment above about a definition of altriciality, because in L. 75 the authors seem to have an implicit definition of altriciality in mind when they say "altriciality might have evolved in earlier hominins", and at this point it is not clear how they will measure or indeed define it. Reading this paragraph (L. 68-82) again after finishing the whole manuscript (and to understand the results, including Extended Data Fig. 3, one needs to read the methods), I infer that the authors use altriciality to mean – in this paragraph at least - a specific and comparatively early stage of neural development at birth, when some but not all neurodevelopmental events have already happened, and referring specifically to Workman et al.'s work. The manuscript would benefit from this second aim to be (a) explicitly defined as a second aim in the Intro, and (b) to

15be explained in more detail (Intro + Results) with regard to the data that are used and what aspect of altriciality they measure, much in the same way that they do for brain and body size proportions for the first aim.

This part of the manuscript has been completely revised according to the feedback provided by the three reviewers. With respect to reviewer #1's concerns, the introduction and results sections now include much more detailed explanations on what these analyses are measuring and how those results feed back into the analyses of extant mammals presented in the first part of the paper. The discussion also incorporates these results much more explicitly now. The new way these results are presented (including one new main figure, one extended data figure and one extended data table) improves understanding of these results.

SPECIFIC / MINOR COMMENTS AND QUESTIONS

** Abstract

L. 21: "Based on our data" rather than "Based on our results".

This has been changed.

L. 26-28: I didn't quite know how to interpret this last sentence. Seeing as how the authors stated their main findings earlier in the abstract, I expected this last sentence to be some final, overarching conclusion but it feels a bit vague instead, as if the authors are not yet divulging how their study shed light on this hypothesis. See also my earlier, general comment on plasticity.

The final sentence of the abstract is now clearer about the implications of our results.

** Introduction

- L. 35: see also Rosenberg 2021 (<https://doi.org/10.1146/annurev-anthro-111819-105454>) for the association with the development of social and emotional relationships.

This reference has been added.

- L. 46-47: better to say that selection favored neonates with smaller heads/brains *at the time of birth*, otherwise it is a bit ambiguous. Rosenberg 1992 is a good reference here but the classical reference is Washburn 1960 (10.1038/scientificamerican0960-62), which the authors may wish to add.

This sentence has been rephrased as suggested, and Washburn's reference has been added.

- L. 51-52: "it can be inferred that [human altriciality] is associated with substantial adaptive advantages, as otherwise selection would likely have acted against it." I encourage a rephrasing, because right now it reads a bit adaptationist, as if altriciality in humans must have had adaptive advantages in its own right, irrespective of the constraints imposed by the pelvis and/or maternal metabolism, but this is hard to prove. It may have had adaptive advantages in its own right (increased plasticity, opportunities for social learning, etc.) but even the constraints imposed by maternal anatomy and/or physiology favors altriciality as a compromise solution between fetal development and obstetric/metabolic sufficiency. Based on what is written, I presume the latter is what the authors mean, but it is ambiguous.

This sentence has been rephrased considering these comments and also the feedback from reviewer #2, who had similar concerns on the adaptationist nature of this paragraph.

L. 55: add "also" to "...could [also] explain why a seemingly..." because these explanations are not mutually exclusive.

Added

L. 60: "To assess how altriciality evolved in humans", I find the "how" here rather vague (and the rest of the sentence does not make it clearer).

This sentence has been rephrased and moved to the methods section as a way to shorten the introduction.

L. 63-64: the authors could cite their existing ref Jones et al. 2009 (panTHERIA database) for body size variation in mammals.

This reference has been added, but, as mentioned above, this paragraph has been moved to methods now.

L. 72: "species-specific ontogenetic series" of what?

This has been removed to avoid possible misunderstandings.

** Results

- In L. 93-94, the authors state that “body proportion at birth appears to be highly constrained across mammals.” Although body size proportions are not the primary focus of the manuscript, this statement is nonetheless a bit misleading/inaccurate based on the authors’ own data/results and the published body of work on body proportion at birth. Their study relies on matching data for neonatal brain and body size proportions across all taxa studied, which limits the potential sample size (and at N=140 the authors have done well) and thus misses out on a lot of published neonatal body size data. The authors claim that, at <25%, body proportions are highly conserved in mammals, but a range of ~25% is arguably not little. They furthermore state there are minor differences in body proportion across clades (L.91-93), but this ignores the variation within clades (visible in Fig. 1b but not explicated by the authors). Also, bats give birth to neonates/fetuses up to a whopping 45% of maternal body size and these data have previously been presented in the context of human evolution, altriciality and pelvic evolution (Grunstra et al. 2019 <https://doi.org/10.1002/ajhb.23227>). In fact, for 2 of the bat species used by the authors, *Nyctalus noctula* and *Plecotus auritus*, the adult body proportions achieved at birth are 19.5% and 25%, resp., based on the PanTHERIA data. However, Extended Data Table 1 shows a maximum value of 9.99% for the category that includes the bats, and this must therefore be wrong or based on inaccurate data (for reference, Kurta & Kunz 1987 Symp. zool. Soc. Lond., no. 57, even report 19.7% and 30.1% for these 2 species). *Please revise.* Most bat species do not have associated neonatal and adult brain size data, precluding their inclusion in this study, but the statement that body size proportions are “highly constrained among mammals” (well, “placentals” because marsupials are another story) I feel is nonetheless incorrect given all of the above. The authors want to highlight the “contrast” (L. 98) with the variation in brain proportion at birth (where variation within orders *is* interpreted), but they don’t need to throw the variation in body size proportions under the proverbial bus in order to make this point.

As mentioned by reviewer #1, our study includes only species with available data for neonatal brain and body size. In that sample, no species shows a body proportion at birth higher than 25%, but we have included now a reference to Grunstra’s paper and a mention to some bat species showing almost 50% of maternal body size at birth. As suggested by reviewer #1, we have also removed the mention to body proportion at birth being ‘highly constrained’, and we have stated simply that the range of variation of body proportion values is narrower than the range of variation of brain proportion values. The values provided in Extended Data Table 1 have been corrected, as there was indeed a typo, plus the body proportion values have been obtained now with respect to maternal body size.

- L. 150-151 & Fig. 3b: the authors state that unlike in all other well-sampled orders, there is no relationship between the evolutionary rates in body and brain proportions in primates, but it looks like there may in fact be a (weak, negative) relationship in this group if it wasn't for humans. Did the authors calculate the correlation in primates without humans? Because outliers, which humans clearly are in this plot, can have strong effects on correlations and regressions, especially if they are extreme values (as humans are here). I would not characterize an entire clade based on the undue influence of one data point.

All correlation analyses have been repeated including and excluding outlier rate values. Both sets of correlations are now presented in Figure 3 and reported in the manuscript. As anticipated by reviewer #1, some of those correlations are mostly driven by outlier rate values, whereas others are robust to the exclusion of outliers.

- Similarly, but in the opposite way, in the next paragraph (L. 153-159), primates are described to have a particularly strong negative correlation between absolute adult brain size and brain altriciality (brain proportion at birth), which also seems due to the influence of the human data point in the top left corner of the relevant panel of Fig. 3b. Of course humans' contribution to the primate pattern is legitimate as they/we are primates, but the authors should be careful when interpreting humans in their primate context, and indeed when interpreting primates in a broader placental mammalian context, when this "context" is particularly strongly influenced by humans themselves. Even just acknowledging the influence of the human data point might be sufficient here.

As above, we acknowledge the influence of those outlier values (not only humans, but also some branches in the other clades), so we have reported correlations with and without outliers.

- L. 172-174: do these correlations exist within any of the clades?

Those rates are not significantly correlated in primates, but they are in some of the other clades. Because our manuscript focuses on humans, we have described those correlation in the manuscript only for primates, but the correlation results obtained for all the other clades are provided in Figure 3.

- section starting with L. 178 & Figs. 4 & 5: I figured out that the neonatal and adult body and brain sizes are analyzed on a natural log (ln, not common log) scale, but this is not mentioned in the text nor in the legends of Figs. 4 and 5.

Yes, we use natural logarithms and this is specified now in the figures.

- L. 181: the authors report the same slope and intercept for the four best-sampled orders here, but in the legend of Fig. 4b they report a significant difference between carnivorans and primates.

The four orders show the same slope and intercept for the relationship between neonatal and adult **body** size. Carnivorans differ from primates in the relationship between neonatal and adult **brain** size, which is the significant difference reported in the legend of Figure 4.

- section L. 178-204: clade-specific slopes and intercepts are compared and outliers within each clade are inspected separately. This is a nice part of the analysis and the findings are interesting and illustrative. However, the authors make no mention of the fact that carnivorans include two rather distinct group of mammals in terms, namely the terrestrial carnivorans and the aquatic pinnipeds (seals). These two groups differ markedly in where they fall on the altriciality-precociality spectrum, including litter size, all of which seems highly relevant to the focus of this paper. Indeed, I think I can see which carnivoran data points are the terrestrial ones and which the pinnipeds just by looking at the plots in Figs. 4 and 5. With regard to the “outlier analysis” in Fig. 5 I think an argument could be made to regard not the bears as outliers, but the pinnipeds as outliers to an otherwise terrestrial carnivoran pattern. It was not clear from the Methods whether the regressions/pANCOVAs are conducted on the entire sample or not, i.e. including or excluding the apparent outliers, but in case it is the latter, then pinnipeds would be equally if not more appropriate “outliers” as they represent a true grade shift within Carnivora. Also, the big cats (*Panthera* sp.) appear to be missing in Fig. 5c and 5d, and at least *Panthera leo* and *P. tigris* would cluster close to the bears, which ostensibly would have an effect on the regression line, confidence/prediction intervals and possibly on who is an outlier. This requires critical re-analysis.

New analyses have been carried out to compare the scaling relationships between neonatal and adult brain and body size between pinnipeds and terrestrial carnivorans. As stated by reviewer #1, these analyses are very interesting, but, because of space limitations (and because the focus of our paper is human evolution), we cannot discuss them in detail. Therefore, we have provided an extended data figure (Extended Data Fig. 3) providing those additional comparisons. For the comparison between pinnipeds and terrestrial mammals, they show significantly different scaling relationships between neonatal and adult body size, but not between neonatal and adult brain size. pANCOVA analyses are carried out in the sample excluding outliers, and outliers are compared with the rest of the sample afterwards. *Panthera* species are now visible in Fig. 5 and Extended Data Fig. 3.

20- Similarly, the toothed whales are nested within artiodactyls (“Cetartiodactyla”) and they are often compared to anthropoid primates for their increased intelligence, learning capacity, cultural behaviors and sociality. Interestingly, the data points (Figs. 4 and 5) suggest that whales as a whole may not deviate from the terrestrial artiodactyl pattern, possibly because all cetartiodactyls tend to give birth to a single, well-developed fetus. I think it could be interesting to point out. By contrast, suiformes are the odd ones out, which is very likely due to the fact that they alone give birth to litters >1 . Again, a super brief sentence explaining this seems worthwhile and makes the most of the authors’ comparative approach.

As above, the comparison between cetaceans and the other artiodactyls has been provided in Extended Data Fig. 3. For the sake of consistency, we have added a similar comparison for each of the other orders: hominids (great apes and humans) with respect to the other primates, and murids with respect to the other rodents. As mentioned by reviewer #1, these comparisons yield the same scaling relationships between these subclades within each order in spite of their clear differences in adult brain size and/or ecological specializations. Because of space limitations, we have not discussed these results in detail, but all the plots and pANCOVA results are provided in Extended Data Fig. 3 and its legend.

- where are the rodents in Fig. 5? Was there no apparent outlier in this group? This should at least be mentioned somewhere.

The original version of the paper did not include rodents in this figure because there are no clear outliers for the neonatal-adult brain size relationship within rodents. However, we agree with reviewer #1 that figure 5 seems to be missing the rodent panels. Therefore, we have identified the species with the largest residual in the neonatal-adult brain size plot, which corresponds to the golden hamster, and we have compared it to the other rodents in the same way we have compared the other outliers with their orders. Interestingly, the golden hamster does show a significant difference in brain size relationships with respect to the other rodents, and a border-line significant difference in body size relationships.

The points below relate to my major/fundamental comment above on the neurodevelopmental timing analysis. For example:

- L. 208-211: this needs more explanation.

As mentioned previously, these analyses have been revamped and multiple clarifications have been added.

- L. 222-225: what are these events? Providing a couple of examples would help.

Examples have been provided.

- L. 231: what “range” are the authors referring to? Event scores > 0.788 or the range 0.670-0.788? And I didn’t understand how “the range of event scores...probably narrower” followed from the preceding part of the sentence about variation in gestation length among hominins. (By the time I reached this part of your interesting manuscript, I felt I had to work too hard to follow the authors.)

This paragraph is complete rephrased now.

- L. 233: “within this range”, is this the same range as above?

This paragraph is complete rephrased now.

- I know NEE employs a format where the Methods only appear at the end, but this section of the results really requires a bit more explanation of these neurodevelopmental events and how the present authors calculated whether they would occur pre- or postnatally.

Some general explanations are now provided within the results section, just before describing the actual results, and more detailed explanations are provided in the methods section.

- a brief explanation of the significance of myelination or its developmental timing here (Results) or in the Discussion would be appreciated.

Additional explanations have been added.

** Discussion

- L. 259-261: “[...] our study has focused on brain development in an attempt to understand whether human altriciality has evolved as a selectively advantageous trait in association with increased brain plasticity.” The authors have not sufficiently explained how brain altriciality leads to increased plasticity, nor have they argued (however briefly) that (and why) neural plasticity is selectively advantageous earlier on in the manuscript, so this sentence comes somewhat as a surprise to the reader. See my one of my earlier general comments.

22Clarifications on the possible relationship between altriciality, brain plasticity and behavioural complexity are now provided in the introduction.

- L. 287-289: I don't understand the conclusion the authors draw here; I don't follow how the sentence relates to the preceding text or what exactly the point is that the authors try to make.

The new version of the manuscript now includes multiple mentions to the way that our analyses can be used to understand whether human gestation length is shorter than expected or not, and whether the gestation lengths that are normally reported in the human altriciality literature as 'necessary' for humans to be born at the same neurodevelopmental stage as other primates are realistic or not. What these lines (together with Extended Data Fig. 5) show is that those gestation lengths (16 months or 21 months) are not possible given the evolutionary framework where primate gestation lengths evolved, as they would require unrealistically high evolutionary rates along the human branch.

- L. 335-339: Also here, as a reader I feel like I'm missing the relevant background and context explained earlier in the manuscript to (a) expect this kind of discussion and (b) to know what to do with this information.

We have linked this discussion more clearly to our results to emphasize the point that there is a quite narrow range of neurodevelopmental events that have moved from the pre- to the postnatal period during hominin evolution.

- L. 341-350: same as above + it gets a bit technical / specific terms that the non-neural expert may not follow (e.g. "activity-dependent myelination", "brainwave rhythms").

We have tried to provide less technical clarifications of some of these terms, but we feel that we cannot not avoid them completely, as this is a paper about brain development (which we have also highlighted in the title now).

- L. 352-360: The relevance of this paragraph is not very clear to me. (Given the above, if the authors need to shorten the manuscript due to limits on article length, this could probably be achieved in this part of the Discussion.)

We think that this paragraph is important because it reinforces the idea that most human-specific behaviours do not emerge during the early postnatal period, but some time afterwards when, according to our results, there are no differences in the pre- or postnatal occurrence of

neurodevelopmental events between humans and other hominins.

** Methods

- L. 438-439: very good.

- L. 442: I presume the authors use Pearson correlations?

Yes, this has been clarified.

- L. 458: Is the expected scaling relationship established on the sample excluding the outlier? I think this should be made explicit. (See also my earlier comment in the Results.)

Yes, this has been clarified.

- L. 501-503: I would rather say “speculations” or something more careful than “general observations” when it comes to the inferences made about Neanderthal gestation length by Trinkaus, because this has since been debunked/refuted and is no longer held to be true by the community.

This sentence has been removed, as this part of the manuscript now uses species-specific gestation lengths estimated after Monson et al 2022 using values from this paper and from DeSilva 2011.

- L. 504-515: the authors describe how they dealt with the uncertainty in gestation length for the different fossil hominins, but maybe they can also quickly report on how the main results/outcomes of this analysis differed (or whether they differed) depending on what gestation length was used for these different species. Right now this is not transparent.

As above. A detailed explanation is now provided in the methods section on how species-specific gestation lengths have been estimated for fossil hominin species, and how error ranges have been obtained.

** Figures and Tables

Fig. 1: Great figure and I appreciate all the original data being displayed alongside the boxplots. Perhaps the authors can consider briefly defining “brain and body proportion” in the legend (e.g.

“% adult size attained at birth”), because at this point in the manuscript the reader is not yet familiar with these terms.

This clarification has been added to the legend of the figure.

Fig. 2: What do the little open and filled (black) circles on the nodes and tips represent, and what does the difference in the circle size represent? Are they scaled to the values of brain and body sizes, proportions, etc.?

Yes, this has been clarified in the legend of the figure.

Fig. 4: units of measurement / log scale missing (see my earlier comment about this).

This has been added.

Fig. 5: Panthera sp. (x3) are missing from plots c and d (see also my earlier comments about this fig and the outliers)

These data points are visible now and they have been explored in more detail in Extended Data Fig. 3

Extended Data Fig. 2: not really sure what the authors are trying to show here, so please clarify.

As explained above, this figure shows that the evolutionary rates that would be required for humans to have a gestation length of 16 months or 21 months are unrealistically high in comparison with the rates observed across mammals.

Extended Data Fig. 3: I found this figure challenging to interpret and understand when reading the manuscript chronologically (see also my earlier comments on general clarification of the aim and analysis of this part of the study). Among other things, I found the combination of gestation length along the Y axis and the slanting of the lines confusing; why display gestation length when you're not depicting any variation among taxa in gestation length? Why are the lines slanted? Some lines cross the vertical birth line, but it is not clear from the fig whether it's significant/meaningful that the top or bottom of the taxonomic lines crosses the vertical birth line. What determines the position of the taxonomic lines along the x axis? Without more clarification in the main text (i.e. before the methods section) and in the figure legend itself, I thought this figure was not very accessible.

This figure has been removed and replaced with the new Figure 6 and Extended Data Figure 4, and necessary explanations have been provided in the legends of these new figures. We believe that the new figures are much clearer and much more complete than the original Extended Data Figure 3.

** Data files and scripts

Looks like all data files and scripts for a reproduction of the analysis are there.

Reviewer #2 (Remarks to the Author):

Thanks very much to the authors for this manuscript and especially for the study that went into it. With major revisions (including additional analyses) it will be a strong contribution to human evolutionary science that will move us forward on the question of altriciality. I have organized my feedback into four categories, below: (1) The problem with the adult sample being both sexes (with no additional analysis using only females), (2) Clarity and consistency of terms/concepts, (3) Additional references, (4) Evolutionary thinking.

Thanks again, very much.

Sincerely,

Holly Dunsworth

1. The problem of the combined sex adult sample

Line 385 reveals that the neonatal measures are being compared to a combined sample of adult males and females, when they should be used in comparison to female adults exclusively--if not as the sole analysis, than as an additional one included in this paper. This is a problem that requires major revision. I'm sitting here worried that I misinterpreted line 385, given the figures have silhouettes of curvy human females (not males), which leads readers to assume all the adults in the analyses are females, only. If that's true, that's fantastic and there is no problem. But the word female is absent entirely from the paper and where "maternal" shows up, it's to describe previous work, not this analysis. And then there's line 385. So, it seems those figures with the curvy human female silhouettes are misrepresenting the actual analysis. The only acceptable analysis to my mind excludes adult males. If argued well, I could imagine how an analysis using two kinds of adult samples (one pooling the sexes and one female-only) could fly, but the analysis in this manuscript that uses only a pooled adult sample does not fly. Maternal/female adult size is the ideal comparison because of what the authors say in line 385 and because of known metabolic constraints of gestation and lactation that are reflected in/correlated with

26maternal size (that the authors acknowledge elsewhere in the paper). New analyses need to be included with a female-only adult sample.

As explained in our manuscript, our paper looks at the evolution of human altriciality from a brain development point of view, not from an evolutionary constraint point of view. For that reason, we did not feel that using maternal body size was more appropriate than using adult body size in general. However, we do acknowledge that using male-female pooled adult body size measurements is noisy because of the different levels of sexual dimorphism in adult body size across mammals, so we have used maternal body size in the revised version of the paper. The new results are similar to those obtained previously, so we have chosen to include in the paper only the maternal body size values for the sake of simplicity.

2. Clarity and consistency of terms/concepts: Altriciality, Brain altriciality, brain size altriciality, brain proportion, body precociality, body proportion, etc.

There's altriciality and then there's what is under study in this paper, which is part of that phenomenon/concept but not equivalent it. Please make all this very clear from the very beginning of the paper, and then use the specific terms about what you're specifically discussing within the overall concept of altriciality throughout.

As explained above, we have included explicit clarifications on what it is understood as altriciality and precociality in the published literature, and on the specific measurements that we use in the different parts of our paper. We have also explained that exploring brain proportions (as % values) is important because of the historical importance of these values in discussions about human altriciality, providing some relevant classic references.

Brain size altriciality needs to be defined in the Introduction, not in the first sentence of the results. And the variable is actually called "brain proportion" elsewhere, so there's really a need for consistency and clarity on this, throughout the manuscript. When used, please call it "brain size altriciality" not just "brain altriciality" because the latter conjures other developmental phenomena beyond mere size, which are acknowledged in the paper and considered separately. Again, this choice of variable has got to be set up in the Intro with a stronger context for what "altriciality" is as well. The manuscript could really use a firmer foundation there.

'Brain altriciality' has been replaced with 'brain size altriciality' throughout the manuscript, and the requested additional explanations have been provided.

Line 136. I believe this is a typo and is meant to say "body precociality" which should be clearly

27defined now because I believe it's the first mention in the text. I'm also suggesting that you call this variable instead "body size precociality" because just "body" alone conjures motoneuronal development which is considered separately. But wait! It's "body proportion" elsewhere. This is the same problem as outlined above.

This sentence has been removed, as it was confusing, and 'body precociality/altriciality' has been replaced with 'body size precociality/altriciality' throughout the manuscript.

3. Three References that need to be cited in collaboration

1. Karen R. Rosenberg. 2021. The Evolution of Human Infancy: Why It Helps to Be Helpless. *Annual Review of Anthropology* 50:1, 423-440

<https://www.annualreviews.org/doi/abs/10.1146/annurev-anthro-111819-105454>

Rosenberg's paper offers many things including (a) a stronger basis for the Introduction of this manuscript and (b) ideas to cite about the long postnatal growth period being adaptive and potentially the target of selection itself (rather than the byproduct of constraints on gestation and pregnancy).

This reference has been added.

2. Holly Dunsworth. 2021. (Chapter 27) There is No Evolutionary "Obstetrical Dilemma". In C. Tomori and S. Han, (Eds.), *The Routledge Handbook of Anthropology and Reproduction*. Taylor and Francis. https://digitalcommons.uri.edu/soc_facpubs/41/

This paper supports the arguments and conclusions made in this manuscript that increasing absolute adult brain size is causing decreasing relative neonatal brain size. From Dunsworth 2021: "It has long been taken for granted that as monkeys, apes, and hominins encephalized, their postnatal brain growth increased and so their relative brain size at birth decreased. No one has ever asserted that increased encephalization should occur during gestational days alone. And yet, this is what Gould's argument implies—that in order to grow larger brains, our ancestors should have reversed primate tradition and accomplished increased brain growth in utero. To be clear, this is an assumption that human encephalization should have been unlike any other primate's. However, if one approaches human encephalization from a comparative primate perspective, one expects that with increasing adult hominin brain size there must have been decreasing relative brain size at birth, regardless of childbirth difficulty." And the paper also goes into the context for why human evolutionary scientists have long assumed or entertained the idea that humans should have a much longer gestation/larger percent of adult brain at birth than we do.

This reference and Gould's reference have been added too.

3. Tesla Monson et al. 2022. Teeth, prenatal growth rates, and the evolution of human-like pregnancy in later Homo. PNAS <https://www.pnas.org/doi/10.1073/pnas.2200689119>

This paper estimates gestation length in fossil hominins and is thus highly relevant to that portion of the manuscript.

This reference has been added and Monson's framework to estimate gestation lengths in fossil hominins has been used as described above and in the methods section.

In addition, I found the fossil hominin portion of the text, but especially the table that goes along with it, a bit hard to follow. Maybe it's because I knew the Monson paper and so I was thrown by what I brought, myself via that paper. But re: the table: The headers on the columns should better explain what those columns contain. The figure doesn't and shouldn't have to do the job for them. And if there is room to convey "all the rest of the hominins" instead of "the rest" that would also help. To a newcomer reader seeing "Prenatal: the rest" is disorienting. I think it's as simple as editing the headers to read: Occurs prenatally, Occurs perinatally, occurs postnatally, or something like that.

The table has been amended as suggested by reviewer #2, but it has been moved to extended data. Replacing this table, a new figure (Figure 6) is included in the main paper. That figure has the same information as the table, but it conveys it in a clearer and visual way.

4. Evolutionary thinking

Line 52. This little bit on evolutionary thinking could use a bit more nuance. As-is it's just not correct. That something observable exists (i.e. has not been selected against and eliminated) does not mean that it therefore confers a substantial adaptive advantage. What works, works. In this case specifically, older children and adults are brilliant at many things and one of them is caring for young with an extended postnatal growth period. It need not experience positive selection directly in order for it to exist.

The introduction has been rephrased to remove the adaptationist bias raised by reviewers #1 and #2. It still discusses the possibility that altriciality has been selected, but it does so in a more nuanced way.

But "moving a greater proportion of brain maturation to the postnatal period" (line 56) is not

29what is happening, biologically. I think it's important (though difficult to articulate) to point this out because it looks like this is being considered as a target of selection, or is being described as others' hypothesized target of selection. It cannot be. Maybe I sound like I'm splitting hairs or arguing semantics but I'm not. If "altriciality" in humans or, instead, what I would call a long postnatal developmental period (and can do so even more confidently thanks to this paper) was adaptive and was being increased in duration by selection, then that need not be accomplished by taking away gestation time, by being born earlier. It could be accomplished by lengthening the postnatal growth period. And if it occurred because of a shortening of gestation, then it's by exactly that, selection for a shorter gestation or an earlier birth, not by "moving a greater proportion of brain maturation to the postnatal period." That "moving..." bit describes how the numbers look to us, but it's not what biology would be literally doing over time. To try to convey what I mean, here's that sentence, reworded to describe the biology, instead: "Indeed, a long postnatal growth period offers the opportunity for ..."

The revised manuscript states even more clearly that the apparent altriciality of humans is not the result of changes related to the prenatal period, but that it results primarily from changes in adult brain size. The impression that a small part of neurodevelopment has been 'moved' to the postnatal period during hominin evolution is supported by results in Figure 6, but we discuss extensively that this is not because of a shortened gestation, but because of the longer scale of neurodevelopment (which is ultimately linked to adult brain size). Regardless, we have rephrased the introduction as suggested by reviewer #2.

At the start of the Discussion, in the first sentence with "humans evolved their high level of brain altriciality at a very fast rate" I had a familiar reaction to one I tried to convey above about "moving...". (note: adding size so that the variable is called "brain size altriciality" would go a long way to clarifying this paper.) Because of the results section, I can't help but assume that you're referring to your variable "brain proportion" here. Talking about humans evolving a proportion of their adult brain size at birth is a problem. That's not an actual trait of an animal. Selection doesn't "act" on the size of a neonatal brain in relation to what it's eventual adult size will be. That's a proxy we made up. That's how the field has talked about it and it's just strange. When I read that sentence I shouldn't have to substitute "long postnatal growth period" or "high rate of gestational growth" or "short gestational growth period" in my head for "high level of brain altriciality" but I am because neonatal/adult brain size at birth is not an actual trait that selection can "see". I hope I'm making sense. Of course relative brain size at birth evolves, but it evolves the way that sex differences in body size evolve: the thing itself is changing over time but not because of selection directly on the thing itself. Absolute brain size at birth, yes. Gestation length and growth rate, yes. Postnatal length and growth rate, sure. The brain

30proportion variable is great for study, but thinking about it like it's an adaptive trait itself is problematic. It's fine being a proxy for long postnatal growth period in the evolutionary analysis. This isn't a criticism of the study itself; it's a critique of the assumptions and thinking that this study fails to support. Maybe my suggestion, then, is that the authors acknowledge this issue and that they're using a proxy variable not an actual trait that can experience selection in their paper. It will help strengthen the case!

We have made it clear that brain proportion is a simple proxy for altriciality, but not a real trait selection can act on. In addition, our results and discussion emphasize now that brain proportion at birth is not significantly correlated with neurodevelopmental stage at birth in primates and that, therefore, it appears to be a particularly poor proxy for neurodevelopmental altriciality, at least in primates.

And it will really help with that hypothetical exercise against Gould's influence on human evolutionary science (see all the context in that Dunsworth 2021 paper above) starting on line 277. That (useful) exercise you undertake shows how ridiculous it has been for our field to assume that percent brain size at birth is an actual trait! But because that's not acknowledged, it's a strange read. I think readers need to know what the point of this exercise is because many are stuck in thinking that percent brain size at birth is something that could be selected and so they elevate it to the level of the traits that can and that muddies this whole endeavor. Most importantly, the summary/conclusion paragraph will be a lot stronger when edited to reflect this issue that your study is revealing to our discipline, which is much needed. In general, not just with this important aspect, this paper really deserves much clearer statement of its fresh approach and fresh insight to the question of human altriciality.

The new clarifications explained above help deal with this issue too. We do make it clear now that brain proportion is not an accurate proxy for altriciality within primates.

Line 98...That's an interesting range of relative neonatal brain size across orders and so it seems like a missed opportunity to consider whether those born with a higher percent of adult brain are more precocial than those born with less. I hope you can add something about that. However, if tracking down those observational data is a challenge, then that may be beyond the scope of this paper and may, instead, be acknowledged in some useful, contributing way.

Because of space limitations, we have not discussed those behavioral differences between species with higher or lower brain proportion values. However, as mentioned above, we have looked at the correlation between brain proportion values and neurodevelopmental stage at birth. Interestingly, a significant correlation exists across the complete mammalian sample, but not

when restricting the comparison to primates.

Paragraph starting at line 98. I don't think the relevance of these results is considered enough in the Discussion. It seems to me that the text should say more explicitly that percent neonatal brain size isn't helpful for understanding human evolution. Right? If so, that's a revelation for the field and needs to be made clear. This links back to my comments about it not being an actual trait, but instead a proxy and in this case... it's the long postnatal period

This point has been discussed explicitly as per my comments above.

Reviewer #3 (Remarks to the Author):

This is an interesting paper and the topic presented is likely to be of interest to a broad audience. However, I have some major concerns that preclude my recommendation for publication which I shall outline below.

The first and primary concern I have is with regards to the methods used to evaluate evolutionary rate heterogeneity. There are several methods available in the literature to do this – and the one the authors choose to use (mvBM) is one that has been criticised (Griffin & Yapuncich 2016, 2017) without clear rebuttal. Further, alternative approaches to estimating trait rate variation systematically outperform mvBM (Cooney and Thomas 2021). The citation history for the method (the authors reference 60) speaks volumes – whilst the method has been used in a few instances, it is largely cited by the authors themselves or in the context of discussing methodological approaches. I expect the authors will fight on this point, but the published criticisms of this approach are difficult to ignore.

We respect the opinion of reviewer #3 regarding the use of mvBM, but we would like to highlight that a rebuttal to Griffin & Yapuncich (2016) has been published (*Smaers JB & Mongle CS 2017. On the accuracy and theoretical underpinnings of the multiple variance Brownian motion approach for estimating variable rates and inferring ancestral states. Biol. J. Linn. Soc. 121, 229–238*). Regardless, we have repeated all our calculations using a reversible jump Brownian motion (rjBM) approach as implemented in BayesTraits software, which is a more commonly used method. Although the individual rate values obtained using both methods differ, the general conclusions regarding human variation with respect to the other species are exactly the same. For the sake of simplicity, we have chosen to present only one set of results, and we have decided to use rjBM-based estimates to ease potential concerns of reviewer #3.

The second major concern I have with the manuscript as it stands is the use of proportions as a

32standalone evolutionary entity. The issue here is that the authors are treating ratios/proportions as evolving quantities, using methods that assume data are distributed according to the predictions of evolutionary models. This is likely to not be the case. Selection does not “see” a proportion, but rather neonatal brain size (or body size, etc). Why not study the trait of interest whilst accounting for the other? The risk here is that by treating these quantities as the target of evolution, it introduces biases in the data – especially when studying traits that are hugely collinear such as brain and body size. By taking advantage of the approaches that allow multivariate models (which are utilized in this manuscript for pattern comparisons) not only does it reduce biases, but it also streamlines the paper.

As explained extensively in our answers to reviewers #1 and #2, understanding the variation and evolution of brain proportion values is important because of the historical importance of this variable in discussions of human altriciality. We acknowledge the issues associated with the study of those proportions though (and we have discussed them more explicitly in the revised manuscript), and this is why the second part of the paper looks at the relationship between neonatal and adult brain and body size in a PGLS regression context.

My final concern lies with the analyses regarding fossil hominins. I’m reticent to even refer to these even really an analysis, rather a set of calculations. It is not really clear what exactly the authors seek to gain from the calculations they make nor what they are trying to say with these sections. It looks to me that they are simply trying to estimate a quantity from fossil hominins (from essentially no data) that they then simply use in a brief descriptive paragraph in the discussion. I would argue that without significantly more statistically rigorous techniques and much more in-depth analysis/discussion, these sections are not adding much – if anything – to the manuscript.

As mentioned previously, we have changed these analyses and we very much hope that reviewer #3 will see their value now. In doing so, not only have we been able to illustrate variation in neurodevelopment during hominin evolution, but these analyses now feed into the analysis of the extant mammalian sample, thus revealing new important patterns of variation. For example, those calculations have allowed us to estimate that the gestation length that would be required for humans to be born at the same neurodevelopmental stage as a chimpanzee is not 16 months or 21 months (as stated in the classic literature), but a mere 10.7 months, which is only slightly above the actual human gestation length of 9 months. This is a completely novel result that strongly undermines the idea that humans are substantially more altricial than expected, and it challenges views that have been held for decades. In addition, these analyses have allowed us to show that there is no significant correlation between % of brain size at birth and neurodevelopmental stage

at birth within primates, which shows how misleading it is to rely on percentage values to discuss human altriciality.

While we have to agree with reviewer #3 that these calculations are based on estimates, we believe that they are still valuable and that they shed light on the evolution of human brain development in a way that no previously published study has. We explicitly discuss in the manuscript the uncertainty associated with these estimates and how we have dealt with this. Although individual values yielded by these analyses are not expected to represent the exact timing of neurodevelopment in fossil hominins, the general trends identified by these analyses do represent real long term trends. For example, our analyses do indicate that some neurodevelopmental events were moved from the prenatal to the postnatal period during hominin evolution, but that the range of events that underwent this evolutionary shift is quite narrow in comparison with the whole scale of neurodevelopment. Our analysis also consistently indicates that it is mostly processes related to myelination that were exposed to this evolutionary shift. We believe that this is valuable information that furthers our understanding of the evolution of human brain development, even if it is based on estimates.

Decision Letter, first revision:

16th August 2023

Dear Dr Gomez-Robles,

Your manuscript entitled "The evolution of human altriciality and brain development in comparative context" has now been seen by the same three reviewers, whose comments are attached. While the reviewers are encouraging and believe the manuscript has progressed well, they still have a number of concerns which will need to be addressed before we can offer publication in Nature Ecology & Evolution. We will therefore need to see your responses to the criticisms raised and to some editorial concerns, along with a revised manuscript, before we can reach a final decision regarding publication.

We therefore invite you to revise your manuscript taking into account all reviewer and editor comments, particularly those of reviewer 3 regarding the study of proportions as evolutionary variables. Please highlight all changes in the manuscript text file.

34When revising your manuscript:

* If you have not done so already please begin to revise your manuscript so that it conforms to our Article format instructions at <http://www.nature.com/natecolevol/info/final-submission>. Refer also to any guidelines provided in this letter.

[REDACTED]

Nature Ecology & Evolution is committed to improving transparency in authorship. As part of our efforts in this direction, we are now requesting that all authors identified as 'corresponding author' on published papers create and link their Open Researcher and Contributor Identifier (ORCID) with their account on the Manuscript Tracking System (MTS), prior to acceptance. ORCID helps the scientific community achieve unambiguous attribution of all scholarly contributions. You can create and link your ORCID from the home page of the MTS by clicking on 'Modify my Springer Nature account'. For more information please visit please visit www.springernature.com/orcid.

[REDACTED]

Reviewer expertise:

as before

Reviewers' comments:

Reviewer #1 (Remarks to the Author):

I would like to thank the authors for taking my previous comments and suggestions into account and incorporating them into their manuscript in a thoughtful manner. I congratulate the authors on producing a very interesting and exciting piece of work. I am happy to recommend this manuscript for publication, although the authors and/or editor may wish to think about revising the following potential minor inconsistencies:

The maximum value for body proportion at birth is listed as 20.63% ("others", Extended Data Table 1) whereas in the Results section it still reports 25% as the upper limit (L. 107) and Fig. 1b also seems to still have the latter as an upper limit. However, both the text and the figure would appear to be at slight odds with the new data based on maternal body mass.

- L.170: why not mention that humans are part of these outlier rates in primates? (Not an inconsistency.)

- L. 245: not "other great apes" but "great apes" (humans are hominids but not great apes).

- L. 360: are humans considered part of the great apes in this sentence ("...remains fairly stable among the great apes.")? (See my point above.)

With very best wishes,
Nicole Grunstra

Reviewer #2 (Remarks to the Author):

Thanks to the authors for these revisions. The Introduction is so much clearer now. And the Discussion from line 305 to the end is superb. The text integrates the powerful figures... powerfully. This paper will make a big and positive impact on the field.

Thanks for pointing out that the proportions (of body and brain at birth compared to adult sizes) are "not exposed to selection per se". I do wonder about the choice that's made in the results and beyond to discuss their change over time as "evolutionary change" and the rate of change as the "evolutionary" rate. I wonder whether that is incompatible with what these proportions are: not traits that undergo biological evolution themselves. I don't have an answer about whether/how to use "evolutionary" in discussing their change and rate of it, but I would like to see the authors explain in the paper (make explicit) what these terms (evolutionary change, evolutionary rate) mean for these

36proportions which aren't evolving traits in the biological evolutionary sense. I noticed that on line 283 the percentage is called "this variable" and I wonder if that's not blurring the line too much. How about "proportion" there instead? A side note about line 284: The sentence that begins "When using this percentage value, ..." has a confusing structure. How does using it (as opposed to something else????) result in the part after the comma? A connection is missing or an insinuation is accidentally made merely by how the sentence is constructed. It all makes sense by line 295 but that sentence could be improved for clarity.

I'm trying to learn how 10.7 months was estimated and I'm going to Extended Data Figure 4a where the text is suggesting that I can presumably do so, but there is no "4a" there as indicated in the text. That typo is not the point, though. I urge the authors to include the equation or the logic that they used to arrive at 321 days/ 10.7 months.

Relatedly, the authors should justify their decision to use 275 days for human gestation even if it's to merely boldly point to the one specific source it's from. Now that we're here on page 26, in the paragraph that starts on line 576, I'm lost on how the gestation length for the LCA was calculated. Extended Data Table 3 is not referred to in the text and that's relevant. It's missing explanation for how the "ancestral values inferred in the Pan-Homo last common ancestral species (LCA)" were... inferred. These processes (or sources where they reside) should be included, please. I'm also intrigued by the characterization of "unrealistically extreme" gestation length estimates for hominins. I think it's fair to request an explanation for why they are "unrealistically extreme" because it's not intuitive, at least not to me. And the table that goes with this portion of the paper (Extended Table 4) should just spell out the species names instead of using those abbreviations. It will make reading the table easier and cut down on text in the caption.

Three minimal editing remarks:

I think the last paragraph could be a bit clearer than this. I think that the first mention of "timing of neurodevelopment" should allude, perhaps parenthetically, to what that is (maybe as briefly put as "based on neural events" or something) so that it's distinguished from the previous discussion the other aspect of the study that it is.

Line 242: I was tripped up by "former," at first, not knowing if it was merely humans or both humans and Neanderthals together. A quick edit would fix that.

Lines 299 and 301 talk about plural studies but both instances cite only one.

Reviewer #3 (Remarks to the Author):

I think the authors have done a very good job of addressing the concerns raised by all reviewers. There are just a few outstanding issues that I think need to be addressed. I will go through the points

37raised in my previous review in turn.

Point 1: Models for studying rate heterogeneity

I am pleased to see that the overall conclusions remain when using the alternative method for studying rate heterogeneity. There are a few parts in the methods that need updating to reflect this change as currently there is some ambiguity:

L456-457: the approach used is no longer that used in the studies cited. These need to be amended.

L457-162: this description pertains to mvBM but not the rates as output by the rjBM variable rates approach. There needs to be a basic description of the rjBM approach used and what the rates mean.

L467: the authors used default priors - but is that appropriate for the analysis at hand? Note also a change in the default priors on variable rates proposals between BayesTraits V3 and BayesTraits V4 (this has caught me out recently) which may affect things.

L487: I think this inclusion is unnecessary - unless the authors are going to discuss the differences between the two models (and this opens up a whole can of worms), I believe they should present the results of only a single model. As it stands, this does not add anything to the manuscript.

Point 2: Studying proportions

The issue of studying proportions as evolutionary variables is discussed in the responses and in the manuscript - though I am still uncomfortable with their use in the rates models.

I appreciate the author's note that it is important to study these proportions in the context of historical analyses and that they compare the results to those using regression residuals. However, the study of regression residuals as data is rife with problems - it may even be worse than studying proportions (see Freckleton 2009, Seven Deadly Sins of Comparative Analysis). Instead, the models the authors use to study the rate of evolution of brain proportions can very easily be used to study the rate of evolution of brain size whilst accounting for another covariate (e.g. body size, adult brain size, etc.). That is, they can study the rate of evolution of relative brain size without ever using proportions or residuals as evolving quantities. See Baker et al 2016, Biological J Linnaean Society for the variable rates regression model. If the authors choose to proceed without modification, I think it is important to explicitly acknowledge the problems associated with studying proportions in the manuscript. In either case, this will actually give weight to the arguments regarding the unsuitability of percent neonatal brain size (etc) for studying human evolution (see specifically the discussion with Reviewer #2).

Point 3: Fossil hominins

I am convinced that these calculations are valuable - the authors have done a good job of presenting these in a way that will make a considerable impact to the field (see above regarding using percentage values).

All in all, I think that this manuscript would be suitable for publication after some minor revisions.

*****END*****

Author Rebuttal, first revision:

Dear editor,

Thanks for the opportunity to revise our manuscript again. We have addressed all the points raised by the three reviewers as described below, and we have added a few additional final clarifications. All the parts of the manuscript that we have added or modified are identified with a different color (green).

We would like to thank again the three reviewers for their constructive feedback.

Reviewer #1 (Remarks to the Author):

I would like to thank the authors for taking my previous comments and suggestions into account and incorporating them into their manuscript in a thoughtful manner. I congratulate the authors on producing a very interesting and exciting piece of work. I am happy to recommend this manuscript for publication, although the authors and/or editor may wish to think about revising the following potential minor inconsistencies:

The maximum value for body proportion at birth is listed as 20.63% ("others", Extended Data Table 1) whereas in the Results section it still reports 25% as the upper limit (L. 107) and Fig. 1b also seems to still have the latter as an upper limit. However, both the text and the figure would appear to be at slight odds with the new data based on maternal body mass.

We have indicated in the text a more accurate maximum value of less than 21% instead of less than 25%, which was used as a rounder maximum.

- L.170: why not mention that humans are part of these outlier rates in primates? (Not an inconsistency.)

This is mentioned now.

- L. 245: not "other great apes" but "great apes" (humans are hominids but not great apes).

'Other' has been removed.

- L. 360: are humans considered part of the great apes in this sentence ("...remains fairly stable among the great apes.")? (See my point above.)

Yes, this is now specified.

Reviewer #2 (Remarks to the Author):

Thanks to the authors for these revisions. The Introduction is so much clearer now. And the Discussion from line 305 to the end is superb. The text integrates the powerful figures... powerfully. This paper will make a big and positive impact on the field.

Thanks for pointing out that the proportions (of body and brain at birth compared to adult sizes) are "not exposed to selection per se". I do wonder about the choice that's made in the results and beyond to discuss their change over time as "evolutionary change" and the rate of change as the "evolutionary" rate. I wonder whether that is incompatible with what these proportions are: not traits that undergo biological evolution themselves. I don't have an answer about whether/how to use "evolutionary" in discussing their change and rate of it, but I would like to see the authors explain in the paper (make explicit) what these terms (evolutionary change, evolutionary rate) mean for these proportions which aren't evolving traits in the biological evolutionary sense.

We have removed the term 'evolutionary change' when talking about proportions, but we keep using the term 'evolutionary rate' throughout the manuscript because this is the simplest term we have to convey the idea of 'how fast individual branches are evolving with respect to a neutral expectation'. Indeed, we have tried using other terms that convey this same idea (e.g., 'accumulated change per branch', 'observed versus expected change', etc), but they make the manuscript much more difficult to read when used throughout, and they can be confusing, as most readers won't be familiar with those terms. Therefore, in order to address the concern of reviewer #2, we have chosen to explain in the beginning of the 'Evolutionary rates' section what those rates are (page 7, lines 139-144). We believe that this will help readers understand what those rates are before reading about how they vary through mammalian evolution. Additional explanations on how the rates are calculated are provided in the 'Methods' section.

I noticed that on line 283 the percentage is called "this variable" and I wonder if that's not blurring the line too much. How about "proportion" there instead?

'Variable' has been changed to 'proportion'.

A side note about line 284: The sentence that begins "When using this percentage value, ..." has a confusing structure. How does using it (as opposed to something else???) result in the part after the

40comma? A connection is missing or an insinuation is accidentally made merely by how the sentence is constructed. It all makes sense by line 295 but that sentence could be improved for clarity.

The structure of this sentence has been changed.

I'm trying to learn how 10.7 months was estimated and I'm going to Extended Data Figure 4a where the text is suggesting that I can presumably do so, but there is no "4a" there as indicated in the text. That typo is not the point, though. I urge the authors to include the equation or the logic that they used to arrive at 321 days/ 10.7 months.

This explanation has been added at the end of the 'Methods' section (page 29, lines 645-652). The typo in the reference to Figure 4a has been amended.

Relatedly, the authors should justify their decision to use 275 days for human gestation even if it's to merely boldly point to the one specific source it's from.

This value comes from PanTHERIA and the corresponding reference has been added.

Now that we're here on page 26, in the paragraph that starts on line 576, I'm lost on how the gestation length for the LCA was calculated. Extended Data Table 3 is not referred to in the text and that's relevant. It's missing explanation for how the "ancestral values inferred in the Pan-Homo last common ancestral species (LCA)" were... inferred. These processes (or sources where they reside) should be included, please.

We have now explained that this ancestral value has been calculated using the same variable rates ancestral reconstruction approach used in the study of evolutionary rates, which is explained in more detail in the rate analysis subsection of the 'Methods' section.

I'm also intrigued by the characterization of "unrealistically extreme" gestation length estimates for hominins. I think it's fair to request an explanation for why they are "unrealistically extreme" because it's not intuitive, at least not to me.

We have now explained that these values are unrealistically extreme in comparison with the values observed in humans and the other great apes.

And the table that goes with this portion of the paper (Extended Table 4) should just spell out the species names instead of using those abbreviations. It will make reading the table easier and cut down on text in the caption.

Species names have been spelt out in this table.

Three minimal editing remarks:

I think the last paragraph could be a bit clearer than this. I think that the first mention of “timing of neurodevelopment” should allude, perhaps parenthetically, to what that is (maybe as briefly put as “based on neural events” or something) so that it’s distinguished from the previous discussion the other aspect of the study that it is.

Some sentences in this paragraph have been rephrased to improve clarity. The parenthetical clarification has been added (page 5, lines 94-100).

Line 242: I was tripped up by “former,” at first, not knowing if it was merely humans or both humans and Neanderthals together. A quick edit would fix that.

It is both, this has been clarified in the manuscript.

Lines 299 and 301 talk about plural studies but both instances cite only one.

We have replaced the word ‘studies’ by ‘analyses’, which does not imply that we are making reference to several papers.

Reviewer #3 (Remarks to the Author):

I think the authors have done a very good job of addressing the concerns raised by all reviewers. There are just a few outstanding issues that I think need to be addressed. I will go through the points raised in my previous review in turn.

Point 1: Models for studying rate heterogeneity

I am pleased to see that the overall conclusions remain when using the alternative method for studying rate heterogeneity. There are a few parts in the methods that need updating to reflect this change as currently there is some ambiguity:

L456-457: the approach used is no longer that used in the studies cited. These need to be amended.

L457-162: this description pertains to mvBM but not the rates as output by the rjBM variable rates approach. There needs to be a basic description of the rjBM approach used and what the rates mean.

L467: the authors used default priors - but is that appropriate for the analysis at hand? Note also a change in the default priors on variable rates proposals between BayesTraits V3 and BayesTraits V4 (this has caught me out recently) which may affect things.

42L487: I think this inclusion is unnecessary - unless the authors are going to discuss the differences between the two models (and this opens up a whole can of worms), I believe they should present the results of only a single model. As it stands, this does not add anything to the manuscript.

We have included clearer explanations on how the rate analysis has been carried out (page 21, lines 474-479; page 22, lines 482-488; page 23, lines 508-516). We have explained that our rate analysis combines BayesTraits' variable rates approach (rjBM) to calculate ancestral values and a comparison between the observed and expected amount of change per branch. We have re-structured this part of the methods section to avoid ambiguity. We have explained what BayesTraits' rates are, including some additional references, and how they are used to calculate ancestral values and accumulated change per branch in our analyses. We have specified the BayesTraits version we have used and we have removed the mention to mvBM.

Point 2: Studying proportions

The issue of studying proportions as evolutionary variables is discussed in the responses and in the manuscript - though I am still uncomfortable with their use in the rates models.

I appreciate the author's note that it is important to study these proportions in the context of historical analyses and that they compare the results to those using regression residuals. However, the study of regression residuals as data is rife with problems - it may even be worse than studying proportions (see Freckleton 2009, Seven Deadly Sins of Comparative Analysis). Instead, the models the authors use to study the rate of evolution of brain proportions can very easily be used to study the rate of evolution of brain size whilst accounting for another covariate (e.g. body size, adult brain size, etc.). That is, they can study the rate of evolution of relative brain size without ever using proportions or residuals as evolving quantities. See Baker et al 2016, Biological J Linnean Society for the variable rates regression model. If the authors choose to proceed without modification, I think it is important to explicitly acknowledge the problems associated with studying proportions in the manuscript. In either case, this will actually give weight to the arguments regarding the unsuitability of percent neonatal brain size (etc) for studying human evolution (see specifically the discussion with Reviewer #2).

We have discussed in more detail the issues that are associated with the study of proportions as evolutionary variables. We have mentioned these issues in the introduction (pages 4, lines 76-79), and we have discussed them in more detail in the methods section (pages 23-24, lines 518-530), providing some relevant references. We have highlighted again the reasons why we think that exploring the variation of this percentage value over the mammalian phylogeny is important, as this helps understand the classic view that humans are exceptionally altricial, and that human altriciality has evolved in a somehow unexpected way.

Point 3: Fossil hominins

43I am convinced that these calculations are valuable - the authors have done a good job of presenting these in a way that will make a considerable impact to the field (see above regarding using percentage values).

We are pleased that reviewer #3 considers these analyses valuable now. We agree that the new way we present these analyses has made the manuscript much stronger, and we are grateful to the three reviewers for highlighting that the initial version of these analyses was not clear enough. As mentioned in our replies to reviewer #2, we have added some additional methodological explanations on how we have estimated the expected gestation length of humans according to the timing of neurodevelopment (page 29, lines 645-652).

Decision Letter, second revision:

8th September 2023

Dear Dr. Gomez-Robles,

Thank you for submitting your revised manuscript "The evolution of human altriciality and brain development in comparative context" (NATECOLEVOL-23030516B). It has now been seen again by the original reviewers and their comments are below. The reviewers find that the paper has improved in revision, and therefore we'll be happy in principle to publish it in Nature Ecology & Evolution, pending minor revisions to satisfy the reviewers' final requests and to comply with our editorial and formatting guidelines.

[REDACTED]

Reviewer #3 (Remarks to the Author):

I am pleased to see this manuscript again; it is very clear to read, and the authors have clearly outlined their rationale behind the approaches taken. The authors have done a good job of clarifying

44the queries I had in my previous report. I therefore am happy to recommend this paper for publication.

Minor edits:

L24-25: "were shifted" rather than "was shifted"

L487: missing a word, should read "conform TO a Brownian motion process"

Our ref: NATECOLEVOL-23030516B

19th September 2023

Dear Dr. Gomez-Robles,

Thank you for your patience as we've prepared the guidelines for final submission of your Nature Ecology & Evolution manuscript, "The evolution of human altriciality and brain development in comparative context" (NATECOLEVOL-23030516B). Please carefully follow the step-by-step instructions provided in the attached file, and add a response in each row of the table to indicate the changes that you have made. Please also check and comment on any additional marked-up edits we have proposed within the text. Ensuring that each point is addressed will help to ensure that your revised manuscript can be swiftly handed over to our production team.

****We would like to start working on your revised paper, with all of the requested files and forms, as soon as possible (preferably within two weeks). Please get in contact with us immediately if you anticipate it taking more than two weeks to submit these revised files.****

In recognition of the time and expertise our reviewers provide to Nature Ecology & Evolution's editorial

45process, we would like to formally acknowledge their contribution to the external peer review of your manuscript entitled "The evolution of human altriciality and brain development in comparative context". For those reviewers who give their assent, we will be publishing their names alongside the published article.

Nature Ecology & Evolution offers a Transparent Peer Review option for new original research manuscripts submitted after December 1st, 2019. As part of this initiative, we encourage our authors to support increased transparency into the peer review process by agreeing to have the reviewer comments, author rebuttal letters, and editorial decision letters published as a Supplementary item. When you submit your final files please clearly state in your cover letter whether or not you would like to participate in this initiative. Please note that failure to state your preference will result in delays in accepting your manuscript for publication.

Cover suggestions

We welcome submissions of artwork for consideration for our cover. For more information, please see our [guide for cover artwork](https://www.nature.com/documents/Nature_covers_author_guide.pdf).

Nature Ecology & Evolution has now transitioned to a unified Rights Collection system which will allow our Author Services team to quickly and easily collect the rights and permissions required to publish your work. Approximately 10 days after your paper is formally accepted, you will receive an email in providing you with a link to complete the grant of rights. If your paper is eligible for Open Access, our Author Services team will also be in touch regarding any additional information that may be required to arrange payment for your article.

Please note that *Nature Ecology & Evolution* is a Transformative Journal (TJ). Authors may publish their research with us through the traditional subscription access route or make their paper immediately open access through payment of an article-processing charge (APC). Authors will not be required to make a final decision about access to their article until it has been accepted. [Find out more about Transformative Journals](https://www.springernature.com/gp/open-research/transformative-journals)

Authors may need to take specific actions to achieve [compliance with funder and institutional open access mandates](https://www.springernature.com/gp/open-research/funding/policy-compliance-faqs). If your research is supported by a funder that requires immediate open access (e.g. according to [Plan S principles](https://www.springernature.com/gp/open-research/plan-s-compliance))

46then you should select the gold OA route, and we will direct you to the compliant route where possible. For authors selecting the subscription publication route, the journal's standard licensing terms will need to be accepted, including <https://www.nature.com/nature-portfolio/editorial-policies/self-archiving-and-license-to-publish>. Those licensing terms will supersede any other terms that the author or any third party may assert apply to any version of the manuscript.

For information regarding our different publishing models please see our <https://www.springernature.com/gp/open-research/transformative-journals> Transformative Journals page. If you have any questions about costs, Open Access requirements, or our legal forms, please contact ASJournals@springernature.com.

[REDACTED]

[REDACTED]

Reviewer #3:

Remarks to the Author:

I am pleased to see this manuscript again; it is very clear to read, and the authors have clearly outlined their rationale behind the approaches taken. The authors have done a good job of clarifying the queries I had in my previous report. I therefore am happy to recommend this paper for publication.

Minor edits:

L24-25: "were shifted" rather than "was shifted"

L487: missing a word, should read "conform TO a Brownian motion process"

Final Decision Letter:

18th October 2023

Dear Dr Gomez-Robles,

We are pleased to inform you that your Article entitled "The evolution of human altriciality and brain development in comparative context", has now been accepted for publication in Nature Ecology & Evolution.

Over the next few weeks, your paper will be copyedited to ensure that it conforms to Nature Ecology

47and Evolution style. Once your paper is typeset, you will receive an email with a link to choose the appropriate publishing options for your paper and our Author Services team will be in touch regarding any additional information that may be required

Due to the importance of these deadlines, we ask you please us know now whether you will be difficult to contact over the next month. If this is the case, we ask you provide us with the contact information (email, phone and fax) of someone who will be able to check the proofs on your behalf, and who will be available to address any last-minute problems . Once your paper has been scheduled for online publication, the Nature press office will be in touch to confirm the details.

Acceptance of your manuscript is conditional on all authors' agreement with our publication policies (see www.nature.com/authors/policies/index.html). In particular your manuscript must not be published elsewhere and there must be no announcement of the work to any media outlet until the publication date (the day on which it is uploaded onto our web site).

Please note that *Nature Ecology & Evolution* is a Transformative Journal (TJ). Authors may publish their research with us through the traditional subscription access route or make their paper immediately open access through payment of an article-processing charge (APC). Authors will not be required to make a final decision about access to their article until it has been accepted. [Find out more about Transformative Journals](https://www.springernature.com/gp/open-research/transformative-journals)

Authors may need to take specific actions to achieve [compliance](https://www.springernature.com/gp/open-research/funding/policy-compliance-faqs) with funder and institutional open access mandates. If your research is supported by a funder that requires immediate open access (e.g. according to [Plan S principles](https://www.springernature.com/gp/open-research/plan-s-compliance)) then you should select the gold OA route, and we will direct you to the compliant route where possible. For authors selecting the subscription publication route, the journal's standard licensing terms will need to be accepted, including [self-archiving-and-license-to-publish](https://www.nature.com/nature-portfolio/editorial-policies/self-archiving-and-license-to-publish). Those licensing terms will supersede any other terms that the author or any third party may assert apply to any version of the manuscript.

An online order form for reprints of your paper is available at a

<https://www.nature.com/reprints/author-reprints.html>><https://www.nature.com/reprints/author-reprints.html>. All co-authors, authors' institutions and authors' funding agencies can order reprints using the form appropriate to their geographical region.

We welcome the submission of potential cover material (including a short caption of around 40 words) related to your manuscript; suggestions should be sent to Nature Ecology & Evolution as electronic files (the image should be 300 dpi at 210 x 297 mm in either TIFF or JPEG format). Please note that such pictures should be selected more for their aesthetic appeal than for their scientific content, and that colour images work better than black and white or grayscale images. Please do not try to design a cover with the Nature Ecology & Evolution logo etc., and please do not submit composites of images related to your work. I am sure you will understand that we cannot make any promise as to whether any of your suggestions might be selected for the cover of the journal.

You can generate the link yourself when you receive your article DOI by entering it here: <http://authors.springernature.com/share>.

[REDACTED]

P.S. Click on the following link if you would like to recommend Nature Ecology & Evolution to your librarian <http://www.nature.com/subscriptions/recommend.html#forms>

** Visit the Springer Nature Editorial and Publishing website at http://editorial-jobs.springernature.com?utm_source=ejp_NEcoE_email&utm_medium=ejp_NEcoE_email&utm_campaign=ejp_NEcoE>[www.springernature.com/editorial-and-publishing-jobs](http://editorial-jobs.springernature.com?utm_source=ejp_NEcoE_email&utm_medium=ejp_NEcoE_email&utm_campaign=ejp_NEcoE) for more information about our career opportunities. If you have any questions please click [here](mailto:editorial.publishing.jobs@springernature.com).**